# FINDING THE THREAD: CONTEXT-DRIVEN INCREMENTAL COMPRESSION FOR MULTI-TURN DIALOGUE GENERATION

## ABSTRACT

Modern conversational agents condition on an ever-growing dialogue history at each turn, incurring redundant re-encoding and attention costs that grow with conversation length. To enhance the efficiency, naive truncation or summarization degrades fidelity, and existing context compressors lack mechanisms for cross-turn memory sharing or revision, causing information loss and compounding errors over long dialogues. We revisit the context compression under conversational dynamics and empirically present its fragility. To address both the efficiency and robustness problems, we introduce Context-Driven Incremental Compression (C-DIC), which treats a conversation as interleaved contextual threads and stores revisable per-thread compression states in a single, compact dialogue memory. At each turn, a lightweight retrieve → revise → write-back loop shares information across turns and corrects stale memories, stabilizing behavior over long term dialogue. A lightweight, *gradient-free* policy is proposed to dynamically manage this memory, adapting on-the-fly as conversational contexts evolve without test-time optimization. In addition, we adapt truncated backpropagation-through-time (TBPTT) to our multi-turn setting, learning cross-turn contextual dependencies without full-history backpropagation. Extensive experiments on long-form dialogue benchmarks demonstrate superior performance and efficiency of C-DIC, supporting a scalable path to high-quality dialogue modeling.

## 1 INTRODUCTION

Conversational agents powered by large language models (LLMs), such as CHATGPT (Microsoft, 2025) and Gemini (Google, 2023), have emerged as ubiquitous interfaces for a wide range of tasks such as brainstorming, code debugging, and data analysis (Yi et al., 2024; Nijkamp et al., 2023). These interactions are characteristically *multi-turn*, where even casual sessions often span dozens of exchanges with topic drifts, cross-turn references, and iterative refinements (Xu et al., 2022). Such interactive adaptability of LLM-based assistants constitutes a pivotal cornerstone of their efficacy and enables capabilities beyond static search or form-based interfaces.

Despite strong single-turn performance, current LLM struggle to manage the dependencies and drift that arise in multi-turn discourse (Laban et al., 2025). The prevalent naive approach, concatenating the entire conversation history to the prompt at every step, introduces two core challenges. First, it induces significant **computational inefficiency**: repeatedly *re-encoding* and *re-attending* to the full dialogue history at each turn incurs high inference-time costs, as self-attention scales quadratically with input length (Vaswani et al., 2023; Tay et al., 2022). Second, it triggers **semantic drift and contextual erosion**: as dialogues evolve, models often *lose the thread*, producing irrelevant responses (Laban et al., 2025). These challenges stem from the model's insufficient focus on dialogue turns that align with the user's evolving intents, especially when such turns lie beyond the model's recency-driven attention scope.

Existing methods address efficiency by truncating history to recent turns (Xu et al., 2022; Laban et al., 2025) or by using a static summaries (Wang et al., 2025; Packer et al., 2024). Truncation discards long-range dependencies, while static summaries tend to be query-agnostic, lossy, and inflex-

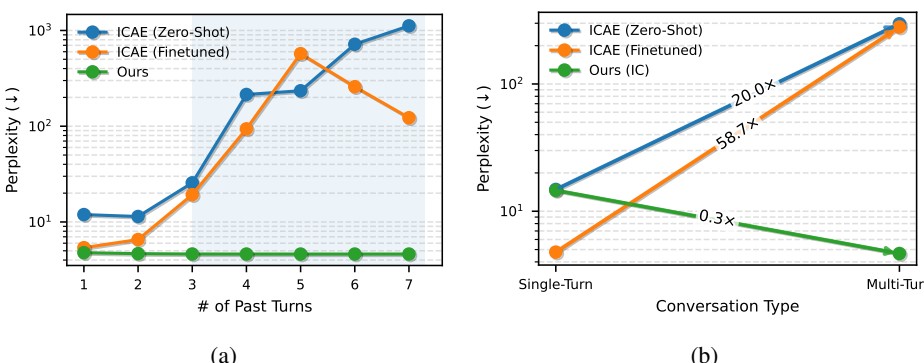

(a)                                                                 (b)

Figure 1: **Static compression collapses under multi-turn conversation; C-DIC remains stable.**
Perplexity ($\downarrow$) for ICAE (zero-shot), ICAE (MSC-tuned), and our method. **(a)** Static baselines rise
sharply after 3–4 consecutive compressions; C-DIC stays flat. **(b)** Moving from single-turn (one-
shot) to multi-turn evaluation, perplexity for static models explodes by *at least* $\sim 1900\%$ while
C-DIC decreases by $\sim 70\%$.

ible for mid-conversation revision (Laban et al., 2025; Ravaut et al., 2024). As a result, truncation-
based methods frequently degrade coherence and adaptability in dynamic multi-turn settings.

On the other hand, a line of work compresses long static documents into a small set of latent vectors
(Chevalier et al., 2023; Ge et al., 2024) for efficiency. However, static, single-shot compressors are
brittle under multi-turn rollout: performance significantly degrades from accumulative compression
across consecutive dialogue turns. This clearly presents the core limitation of the static compressors
lacking mechanisms for memory revision and sharing across consecutive turns.

To address these limitations, we take a principled approach: progressive, topic-aware inference-time
compression of the dialogue history, thereby preserving both efficiency and coherence in multi-turn
dialogues. In particular, agents should retrieve and reason over context that is semantically aligned
with the topic of current, regardless of its position in the history. In absence of such topic sensi-
tivity, even compressed inputs risk overlooking essential context, yielding incoherent or irrelevant
responses.

We implement the above principled approach by introducing **Context-Driven Incremental Com-
pression (C-DIC)**, a framework that treats dialogue as interleaved contextual threads and maintains
a single, compact dialogue memory that stores *revisable per-thread compression states*. Rather than
brute-force full-context prompting, C-DIC runs a lightweight *retrieve → revise → write-back* loop
at each turn, enabling cross-turn sharing and correction as the conversation evolves; *training mirrors
this loop via turn-level, retrieval-aware truncated backpropagation through time over consecutive
same-thread turns*, avoiding full-history backpropagation. Unlike prior compression methods built
for static inputs (Chevalier et al., 2023; Ge et al., 2024), C-DIC enables incremental compression so
that we can continually update the contextual threads with incoming interactions.

C-DIC is grounded in three design principles: **(i) Thread-aware memory retrieval.** At each turn,
the model dynamically retrieves the subset of compressed history relevant to the active thread, ir-
respective of position in the history. **(ii) Incremental compression.** It compresses the current turn
with its thread states, allowing future turns to reuse it without re-encoding the full history. **(iii)
Gradient-free memory update.** To accommodate evolving and revisited topics, C-DIC performs
memory updates online without inference-time gradients. By integrating these components into a
single, *topic-sensitive* framework, C-DIC yields dialogue agents that are both efficient and *contex-
tually fluent*: they retain what matters, discard what does not, and stay aligned with evolving user
intent. Our contributions are as follows:

- We demonstrate that static latent compressors are brittle under conversational dynamics, degrading
  across consecutive compressions and collapsing under turn-by-turn rollout.

- We introduce Context-Driven Incremental Compression (C-DIC), the first framework for turn-
  level incremental compression within a single compact dialogue memory; a *retrieve → revise →*

*write-back* scheme with retrieval-aware TBPTT enables cross-turn sharing and correction, yielding long-range behavior without full-history re-encoding or backpropagation.

- C-DIC improves long-range coherence and reference fidelity while greatly reducing inference cost and input size, outperforming truncation, summarization, and static compression baselines.

# 2 PRELIMINARIES & RELATED WORK

## 2.1 MULTI-TURN DIALOGUE GENERATION

Multi-turn interaction equips conversational agents with the ability to sustain coherent, goal-oriented discourse. By exploiting the entire conversational record, the model can resolve coreference, follow user preferences, and revise assumptions—capabilities that single–turn systems cannot provide. Such continuity is indispensable in real-world scenarios, where users expect the system to remember and adapt to prior turns.

Formally, let a dialogue with $T$ turn be a sequence $\mathcal{D}_{1:T} = \{(q_1, r_1), \ldots, (q_T, r_T)\}$, where $q_t$ is the user query and $r_t$ the system response at turn $t$. At each turn, the large language model (LLM) receives the current input pair $(\mathcal{D}_{<t}, q_t, r_t)$, where $\mathcal{D}_{<t}$ is the entire history up to turn $t - 1$. The training maximises the conditional log-likelihood:

$$\log p_\theta(r_t \mid q_t, \mathcal{D}_{<t}). \tag{1}$$

Crucially, the full history $\mathcal{D}_{<t}$ must be supplied at *every* subsequent turn since each future prediction depends on it. If each exchange contributes on average $L$ tokens, the prompt length at turn $t$ is $tL$. Under typical module such as vanilla self–attention, the cumulative cost over an $T$-turn dialogue is $O(T^3 L^2)$ (Tay et al., 2022). This cubic growth rapidly dominates latency and energy budgets, and empirical studies confirm marked accuracy drops once single-turn benchmarks are converted to multi-turn chat (Laban et al., 2025). We further discuss on key-value caching in the Appendix A.

## 2.2 TEXT-BASED CONTEXT MANAGEMENT

Text-based approximations of the dialogue history are common strategies for mitigating the inefficiencies of full-context encoding in multi-turn dialogues. The simplest approach is truncation, where only the most recent $k$ utterances from the history are retained (Xu et al., 2022; Laban et al., 2025). While effective in limiting input size, truncation often eliminates earlier turns that may contain crucial information. To retain more of the dialogue semantics, summarization-based methods compress the dialogue history into a shorter textual form (Wang et al., 2025; Packer et al., 2024). However, these methods are static summaries, which become outdated as the conversation evolves, leading to inconsistencies or omissions. Static summaries can become outdated as the conversation evolves, leading to inconsistencies or omissions.

## 2.3 CONTEXT COMPRESSION

To move beyond text proxies, recent work proposes context compression, which maps a variable-length context to a fixed set of latent vectors (Wingate et al., 2022; Mu et al., 2024; Chevalier et al., 2023; Ge et al., 2024). AutoCompressor (Chevalier et al., 2023) recursively *accumulates* compression embeddings over dialogue segments; the fine-tuned generator consumes these fixed embeddings without per-turn rewrite. ICAE (Ge et al., 2024) uses a modular autoencoder: a pretrained compressor encodes the full context into a fixed latent matrix $\mathbf{Z}$ consumed by a frozen generator—also *one-shot* and *static* unless the entire input is recompressed. As a common setting, each method appends $k$ trainable compression tokens $C \in \mathbb{R}^{k \times d}$ to the input sequence and runs the language model once. The hidden states at those positions are kept as a dense matrix $\mathbf{Z} \in \mathbb{R}^{k \times d}$ that replaces the raw text in subsequent computation. In this setting, inference cost grows with the constant $k$ rather than with context length, while the latent vectors preserve far more information than truncation or summarization.

Most existing compressors, however, were designed for static documents or single-shot prompts; they cannot incrementally insert or refine compressed context as a conversation evolves, and a fixed $k$ forces an ever-longer dialogue into the same capacity, increasing the risk of forgetting.

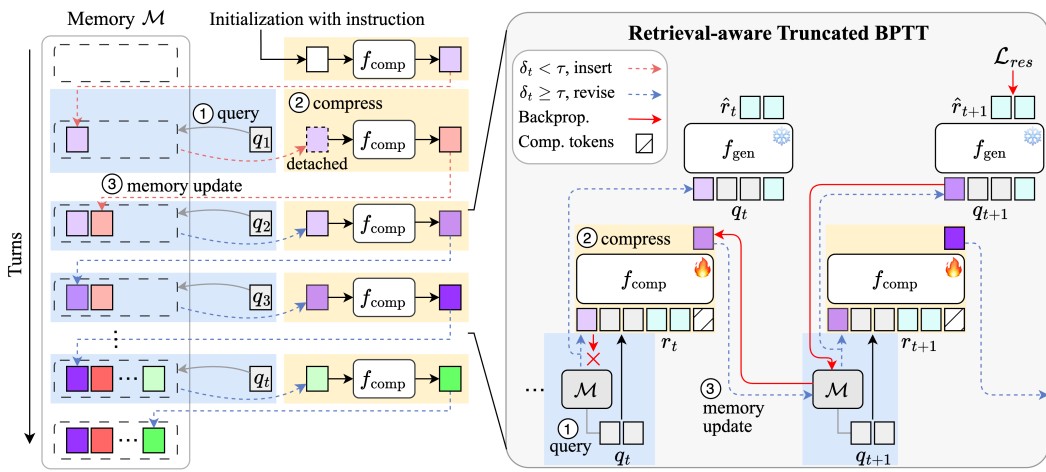

Figure 2: **Overview of retrieval-conditioned incremental compression and ra-TBPTT.** *Left*: We maintain memories $\mathcal{M}$ that store a set of compressed thread states that evolve turn by turn. The memories are initialized by compressing the instruction into a thread state. Each turn then follows three steps. **(1) Query**: given $q_t$, we score the existing thread states and retrieve the relevant states; if the best matching thread is topically irrelevant, we still fetch it for continuity but *detach* it. **(2) Compress**: the trainable compressor $f_{\text{comp}}$ summarizes the retrieved states and the current turn into a new *thread state*, detailed on right. **(3) Memory update**: we apply a gradient-free memory update rule using the peak similarity $\delta_t$ (red dashed = *insert* (topic shift), blue dashed = *revise* the best-matching state (on-topic)); see 3.2 for details. *Right*: For the training of $f_{\text{comp}}$, a per-turn response loss $\ell_t$ is minimized, enhanced with a retrieval-aware TBPTT: gradients flow *one hop* along the argmax usage edge and are truncated ($\rightarrow \times$) thereafter. At turn $t$, the compressor $f_{\text{comp}}$ summarizes $(\mathcal{R}_t, q_t, r_t, C)$ into a new state and writes it back; the frozen response generator $f_{\text{gen}}$ conditions on $\mathcal{R}_t$ to produce $\hat{r}_t$.

Even though Rae et al. (2019); Bulatov et al. (2022); Chevalier et al. (2023) address long-context modeling by augmenting the Transformer with additional memory (compressed past segments, recurrent memory slots, or accumulated embeddings), they still operate primarily at the token or segment level. In contrast, C-DIC introduces an *external*, dialogue-level memory that is updated turn by turn via a retrieval–revise–write-back loop. This design allows compressed states to be incrementally refined and re-used across hundreds of turns, without repeatedly re-encoding the full history or modifying the internal architecture of the base language model.

## 3 METHODOLOGY: CONTEXT-DRIVEN INCREMENTAL COMPRESSION

Recall that conventional prompting strategies suffer from inefficiency issue and fail to provide contextually grounded responses. To handle these issues, we propose **Context-Driven Incremental Compression (C-DIC)**, a framework for scalable multi-turn dialogue modeling that enables efficient, context-sensitive reuse of prior interactions, as illustrated in Figure 2. We model a dialogue as interleaved contextual threads and maintain a compact memory whose slots store revisable, per-thread compressed states. At each turn, the system execute a light retrieve → compress → write-back loop that circumvents the repeated encoding of entire history while allowing cross-turn sharing and correction. While we freeze the response generator, we optimize only the compressor (initialized from ICAE (Ge et al., 2024)) and learnable compression tokens during training. We further discuss this training setup and the rationale for this design in Appendix B.

### 3.1 COMPRESSOR INITIALIZATION

Instead of training a compressor from scratch, we initialize the compressor with a pretrained checkpoint of ICAE (Ge et al., 2024), which was trained on large-scale corpora for one-shot document

compression. We adapt this initialized compressor to our incremental, retrieval-conditioned setting with a frozen response generator. This approach leverages a massive pretrained knowledge of ICAE, endowing our compressor with high-capacity, context-faithful representations without incurring additional pretraining cost.

## 3.2 INCREMENTAL COMPRESSION & CONTEXT-AWARE RETRIEVAL

Our design targets three needs: efficiency over long histories, coherence within an active thread, and learning that concentrates supervision on the memory states the model actually retrieves. Instead of re-encoding the full, ever-growing history at every turn, we maintain a compact memory $\mathcal{M}_{<t} = \{\mathbf{Z}_i\}$ of compressed thread states that evolve with ongoing dialogue. At turn $t$, we (i) **retrieve** a small, query-related subset $\mathcal{R}_t \subset \mathcal{M}_{<t}$ for conditioning, (ii) **generate** the response with a frozen decoder, and (iii) **compress** the new turn into an updated memory slot via a gradient-free write-back policy.

**Turn-wise compression (base case).** We first describe compression without retrieval to fix notation and the learning signal. Given the input pair $(q_t, r_t)$, the compressor produces a compact summary

$$\mathbf{Z}_t = f_{\text{comp}}([\text{Emb}(q_t); \text{Emb}(r_t); \mathbf{C}]; \theta), \tag{2}$$

where $q_t$ and $r_t$ are the turn-$t$ query and response sequences, $\mathbf{C} \in \mathbb{R}^{n \times d}$ is embedding of learnable compression tokens, $\theta$ are the compressor parameters, and $\mathbf{Z}_t \in \mathbb{R}^{n \times d}$ is the resulting compressed state. The frozen response generator generates $\hat{r}_{t+1} = f_{\text{gen}}([\mathbf{Z}_t; \text{Emb}(q_{t+1})]; \phi)$ during training only to calculate the per-turn loss; the generator's parameters remain fixed, so learning concentrates on producing compressed contexts that are useful when consulted. This base case already yields a bounded per-turn cost and a well-defined supervision signal, but it treats all prior context symmetrically and cannot adapt granularity to what the model actually reuses.

**From turn-wise to retrieval-based thread compression** To make compression conditional on the context that actually matters at turn $t$, we introduce a retrieved support set $\mathcal{R}_t \subset \mathcal{M}_{<t}$. Each slot $\mathbf{Z}_i$ is scored by a semantic match with mild recency decay:

$$S(q_t, \mathbf{Z}_i) = \frac{\langle \psi(f_{\text{comp}}(q_t, C)), \, \psi(\mathbf{Z}_i) \rangle}{\|\psi(f_{\text{comp}}(q_t, C))\| \, \|\psi(\mathbf{Z}_i)\|} \, e^{-\alpha \Delta t_i}, \qquad \mathcal{R}_t = \{\mathbf{Z}_i : \, S(q_t, \mathbf{Z}_i) > \tau\}. \tag{3}$$

Here $\psi(\cdot)$ is a pooling function (e.g. mean or CLS token) over token-level representations, $\Delta t_i$ is the number of turns since $\mathbf{Z}_i$ was last retrieved, and $\alpha$ is decay rate, and $\tau$ is a fixed retrieval threshold on the similarity score $S$. If no slot exceeds $\tau$, we fall back to the single best match $\{\mathbf{Z}_{\arg\max_i S(q_t, \mathbf{Z}_i)}\}$. The response generator conditions on the retrieved supports rather than the entire history:

$$\hat{r}_t = f_{\text{gen}}([\mathcal{R}_t; \text{Emb}(q_t)]; \phi). \tag{4}$$

Crucially, compression becomes *incremental* with respect to these supports:

$$\mathbf{Z}_t = f_{\text{comp}}([\mathcal{R}_t; \text{Emb}(q_t); \text{Emb}(r_t); \mathbf{C}]; \theta). \tag{5}$$

This retrieval conditioning focuses $\mathbf{Z}_t$ on the **active thread**, improving long-horizon coherence while keeping per-turn computation proportional to $|\mathcal{R}_t|$ rather than the dialogue length.

**Write-back and thread continuity** To keep the memory both compact and faithful to the evolving topic, we define a *deterministic, gradient-free* update rule. At turn $t$, score the current query against existing slots

$$\delta_t = \max_i S(q_t, \mathbf{Z}_i), \qquad j_t = \arg\max_i S(q_t, \mathbf{Z}_i),$$

and update the memory by either inserting a new state (topic shift) or revising the most similar state (thread continuation):

$$\mathcal{M}_{<t+1} = \begin{cases} \mathcal{M}_{<t} \cup \{\mathbf{Z}_t\}, & \text{if } \delta_t < \tau, \\ (\mathcal{M}_{<t} \setminus \{\mathbf{Z}_j\}) \cup \{\mathbf{Z}_t\}, & \text{otherwise.} \end{cases} \tag{6}$$

Here $\tau$ is the retrieval threshold used in Section 3.2. This policy preserves *thread continuity* by updating the best-matching slot when relevant, and by opening a new slot when relevance is low.

---

**Algorithm 1:** Inference: Retrieve $\to$ Generate $\to$ Compress $\to$ WriteBack

---

**Input:** compressor $f_{\text{comp}}$; frozen generator $f_{\text{gen}}$; tokens $C$; threshold $\tau$; decay $\alpha$

**Output:** $\{\hat{r}_t\}_{t=1}^T$

1   $\mathcal{M} \leftarrow \varnothing$

2   **for** $t = 1$ **to** $T$ **do**

3     **if** $\max_i s_i \geq \tau$ **then**

4       $\mathcal{R}_t \leftarrow \{\mathbf{Z}_i : s_i > \tau\}$   // (1) Similarity-based Retrieval

5     **else if** $\mathcal{M} \neq \varnothing$ **then**

6       $j \leftarrow \arg\max_i s_i; \mathcal{R}_t \leftarrow \{\mathbf{Z}_j\}$

7     **else**

8       $\mathcal{R}_t \leftarrow \varnothing$

9     $\hat{r}_t \leftarrow f_{\text{gen}}([\mathcal{R}_t; q_t]); \mathbf{Z}_t \leftarrow f_{\text{comp}}(\mathcal{R}_t, q_t, \hat{r}_t, C)$   // (2) Generate & Compress

10

11     **if** $\mathcal{M} = \varnothing$ **then**

12       $\mathcal{M} \leftarrow \{\mathbf{Z}_t\}$

13     **else if** $\max_i s_i < \tau$ **then**

14       $\mathcal{M} \leftarrow \mathcal{M} \cup \{\mathbf{Z}_t\}$

15     **else**

16       $\mathcal{M} \leftarrow (\mathcal{M} \setminus \{\mathbf{Z}_j\}) \cup \{\mathbf{Z}_t\}$   // (3) Memory Update

17     update recency counters $\Delta t_i$ for $\mathbf{Z}_i \in \mathcal{M}$

---

Because no gradients flow through selection or write-back, the per-turn cost is small and independent of dialogue length; yet the memory remains semantically coherent, avoiding redundancy and drift without expensive gradient-based editing at inference. We provide a more detailed discussion of alternative memory update variants in Appendix C.

**Retrieval-aware truncated BPTT**   In multi-turn dialogue with compressed memory, the model does not attend to the full history; it consults a tiny set of states selected by retrieval. Standard BPTT (Werbos, 1990) backpropagates through *all* past turns (costly and misaligned with usage), while conventional TBPTT (Schmidt, 2019) truncates by a fixed window (agnostic to which turns were actually consulted). We therefore align credit assignment with *retrieval-defined* dependencies.

Specifically, we minimize the per-turn negative log-likelihood

$$\mathcal{L} = \frac{1}{T} \sum_{t=1}^T \ell_t, \qquad \ell_t = -\log P_\phi(r_t \,|\, q_t, \mathcal{R}_t), \tag{7}$$

and perform a reverse-time backward pass with a *one-hop* truncation along the memory updated chain:

$$\frac{\partial \ell_t}{\partial \mathbf{Z}_{j_t}} \neq 0 \text{ iff } \delta_t \geq \tau, \qquad \frac{\partial \ell_t}{\partial \mathbf{Z}_s} = 0 \text{ for all } s \neq j_t. \tag{8}$$

Equivalently, with a mask $M_{s,t} = \mathbb{1}[s = j_t] \cdot \mathbb{1}[\delta_t \geq \tau]$,

$$\frac{\partial \mathcal{L}}{\partial \mathbf{Z}_s} = \sum_{t=1}^T M_{s,t} \frac{\partial \ell_t}{\partial \mathbf{Z}_s}.$$

For off-topic turns ($\delta_t < \tau$) we keep the arg-max slot for *forward* continuity but detach it in training,

$$\tilde{\mathbf{Z}}_{j_t} = \text{stopgrad}(\mathbf{Z}_{j_t}), \qquad \mathbb{1}[\delta_t \geq \tau] = 0 \Rightarrow \frac{\partial \ell_t}{\partial \mathbf{Z}_{j_t}} = 0,$$

so credit never flows into mismatched memory. Compared to full BPTT (credit through the entire history) and windowed TBPTT (credit through a fixed span), (8) aligns supervision with the *actual causal path* used at turn $t$ (the single retrieved/updated thread), prevents spurious long-range gradients, and scales with the number of *consulted* states rather than dialogue length, exactly matching the retrieval-based, thread-centric structure of multi-turn conversations. We further discuss limitations of full BPTT and fixed window TBPTT in Appendix D.

**Inference** Algorithm 1 outlines the inference procedure. Each turn we retrieve the most relevant thread states and, together with the user query, feed them to the frozen generator to produce a response. We then compress the turn into a new thread state and update memory (insert on topic shift, otherwise revise the active state); inference is fully gradient-free, keeping latency low.

## 4 EXPERIMENTS

### 4.1 DATASETS

To measure long-horizon coherence, we follow the setting of existing works (Xu et al., 2022) and evaluate on Multi-Session Chat (MSC) (Xu et al., 2022) and REALTALK (Lee et al., 2025), two recent multi-session corpora structured around re-engagements occurring after hours or days. All datasets used are publicly available for research purposes.

MSC contains human-human conversations spanning up to five sessions. We use the official training split with 1 001 episodes (averaging 53.3 utterances). For evaluation, we leverage sessions 2–5, yielding an average of 66 utterances per conversation.

REALTALK is a real-world WhatsApp-style corpus featuring 10 conversations collected across 21 days, averaging 21.9 sessions and 894.4 utterances per conversation. To effectively validate the robustness and transferability to longer context, we evaluate on REALTALK in a zero-shot setting. Also, we use two evaluation settings for REALTALK: *all-sessions*, which includes cross-session history to test long-term context, and *per-session*, which restricts inputs to the current session only.

To assess whether MSC and REALTALK genuinely require long-range context and whether target responses are generic, we provide an LLM-based dataset characterization with human verification in Appendix E.

### 4.2 BASELINES

We compare against strong, widely used baselines under the same evaluation protocol and backbone; implementation specifics are described in Appendix B.

- **Full-prompting** feeds the entire dialogue history at every turn.
- **Truncation** uses only the last $k=5$ turns.
- **Summarization** generates recursive textual summaries of the history using a frozen LLM, and use the summaries as history.
- **In-Session RAG** retrieves top-5 prior turns from the same dialogue by semantic similarity and concatenates them for the response generation.
- **AutoCompressor** (Chevalier et al., 2023) splits the history into chunks and recursively compresses each chunk into learnable compression tokens, accumulating a summary.
- **ICAE** (Ge et al., 2024) employs an autoencoder compressor with a frozen generator. We evaluate three update rules to cover common usages:
  - **ICAE (incremental)**: reuse previous latents and re-encode the new turn with them (relay update).
  - **ICAE (one-shot)**: re-encode the full available context each turn (original setting).
  - **ICAE (append)**: compress the new turn and concatenate latents without revision (growing latent length).

All baselines share the same frozen `Llama-2-Chat-7B` (Touvron et al., 2023) generator unless the cited method requires fine-tuning (e.g., Chevalier et al., 2023); this keeps comparisons focused on *context management* rather than decoder capacity.

### 4.3 EVALUATION

We adopt a set of widely used, complementary metrics aligned with common practices in dialogue and summarization research. Following prior works (Chevalier et al., 2023; Ge et al., 2024), we

Table 1: **Main Results on MSC and REALTALK.** We report perplexity (PPL), BLEU, and ROUGE (R-L, R-1, R-2), where REALTALK results are zero-shot. On REALTALK, we use the *per-session* setting due to GPU memory limits of several compression baselines (AutoCompressor and ICAE variants). For text-only baselines (Full prompting, Truncation, Summarization and In-Session RAG), each input context is truncated to the model's maximum context length. Our model clearly outperforms all baseline models in all metrics on both benchmarks.

| Models | MSC | | | | | REALTALK | | | | |
|---|---|---|---|---|---|---|---|---|---|---|
| | PPL↓ | BLEU↑ | R-L↑ | R-1↑ | R-2↑ | PPL↓ | BLEU↑ | R-L↑ | R-1↑ | R-2↑ |
| Full prompting | 41.245 | 0.008 | 0.110 | 0.157 | 0.015 | 25.546 | 0.022 | 0.110 | 0.160 | 0.020 |
| Truncation | 30.890 | 0.012 | 0.128 | 0.184 | 0.024 | 23.830 | 0.023 | 0.114 | 0.165 | 0.022 |
| Summarization | 41.849 | 0.013 | 0.128 | 0.172 | 0.024 | 26.087 | 0.023 | 0.114 | 0.168 | 0.023 |
| In-Session RAG | 35.530 | 0.008 | 0.110 | 0.148 | 0.014 | 26.789 | 0.020 | 0.103 | 0.151 | 0.015 |
| AutoCompressor | 9.285 | 0.012 | 0.121 | 0.145 | 0.019 | 12.625 | 0.019 | 0.055 | 0.134 | 0.019 |
| ICAE (incremental) | 513.774 | 0.006 | 0.057 | 0.069 | 0.005 | 124.024 | 0.020 | 0.068 | 0.086 | 0.013 |
| ICAE (one-shot) | 27.656 | 0.017 | 0.133 | 0.190 | 0.027 | 21.390 | 0.025 | 0.118 | 0.166 | 0.026 |
| ICAE (append) | *Out of memory* | | | | | *Out of memory* | | | | |
| Ours | **8.431** | **0.023** | **0.160** | **0.205** | **0.037** | **9.789** | **0.035** | **0.134** | **0.176** | **0.030** |

Table 2: **Closed-loop vs. teacher-forcing on the REALTALK (all-sessions).** Results for our method under teacher forcing (ground-truth history) and closed-loop generation (conditioning on the model's own past responses) over the full multi-session history.

| Setting | PPL↓ | BLEU↑ | R-L↑ | R-1↑ | R-2↑ |
|---|---|---|---|---|---|
| Teacher-forcing | **9.556** | **0.036** | **0.140** | **0.184** | **0.035** |
| Closed-loop | 9.576 | **0.036** | 0.137 | 0.182 | 0.032 |

employ perplexity (PPL) to measure the generative fluency. For measuring content alignment with human responses, we report BLEU (Papineni et al., 2002) (up to 4-gram precision) and ROUGE (Lin, 2004) as standard reference-based metrics. BLEU computes n-gram precision against reference responses. R-1 (ROUGE-1) and R-2 (ROUGE-2) measure unigram and bigram recall, indicating lexical coverage, while R-L (ROUGE-L) utilizes the longest common subsequence to reflect structural similarity and fluency.

### 4.4 RESULTS

Table 1 presents a comprehensive evaluation of our framework on the MSC and REALTALK datasets. Our method consistently outperforms all baselines across perplexity (PPL), BLEU, and ROUGE metrics, while operating on significantly fewer tokens.

Unlike full-prompting, truncation, or summarization baselines that process raw text linearly, our approach incrementally compresses dialogue turns into fixed-size latent representations and retrieves only the memories relevant to the current query. This retrieval-aware, token-efficient design allows our model to maintain high response quality even as dialogue length increases — achieving superior fluency and coherence while processing less than 0.009% of the raw context on REALTALK (e.g., 8.5k vs. 412k tokens).

Compared to prior latent compression approaches such as AutoCompressor and ICAE, which are designed for static, one-shot settings, our model achieves superior performance with similar or lower memory usage. This demonstrates the effectiveness of our incremental compression strategy and retrieval-based memory refinement in the evolving dialogue. In particular, the naive incremental ICAE baseline catastrophically fails (PPL $\approx 513$). This is due to a structural mismatch between ICAE's one-shot training objective and repeated compression, which we further discuss in Appendix F.

Finally, Table 2 reports our REALTALK *all-sessions* results under both teacher forcing and closed-loop generation; most baselines cannot be evaluated in this setting without truncation due to GPU memory limits. Notably, our model generalizes robustly to REALTALK, a much longer, more open-

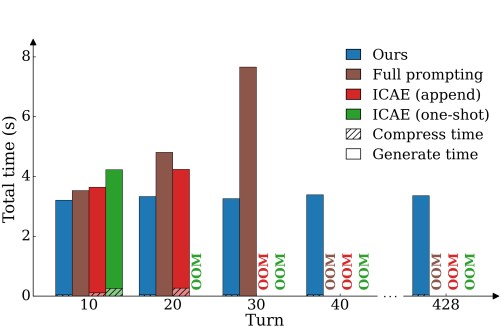
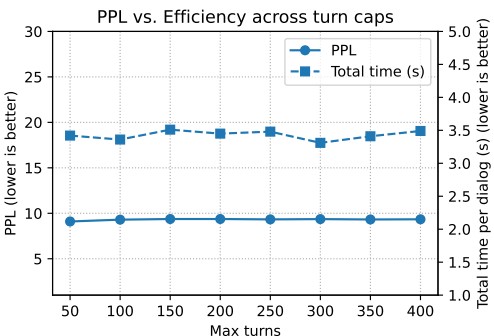

Figure 3: **Latency vs. dialogue length.** Total wall-clock time (s) to process a single dialogue when the maximum number of turns is capped at $\{10, 20, 30, 40, 428\}$. Bars compare **Ours**, Full prompting, ICAE (append), and ICAE (one-shot). The hatched segment denotes *compression time* and the solid segment denotes *generation* time; "OOM" marks methods that run out of memory at that turn. Evaluations use REALTALK in the *all-sessions* setting by truncating to the most recent turns.

Figure 4: **Latency vs. performance across turn caps.** Dual–axis plot with *PPL* (left, lower is better) and *total time per dialog* in seconds (right, lower is better) versus the maximum number of turns $\{50, 100, 150, 200, 250, 300, 350, 400\}$. Evaluations use the all-sessions REALTALK setting by truncating to the most recent turns.

domain dataset, despite being trained only on MSC. As a zero-shot evaluation, this result underscores our method's adaptability on both domain and length shifts. Moreover, the small difference between teacher forcing and closed-loop suggests stable long-horizon behavior over hundreds of turns. We further report mean±std over three random seeds with seed-level significance tests in Appendix G, and additional closed-loop comparisons on REALTALK in Appendix H.

### 4.5 LATENCY COMPARISON

Figure 3 and 4 report total wall-clock time per dialogue as we cap the maximum number of retained turns at $\{10, 20, 30, 40, 428\}$. We decompose runtime into *compression* (hatched) and *generation* (solid). Our method remains nearly constant at $\sim$ 3–3.5s across turn caps, with negligible compression overhead (hatched segment is a thin sliver). In contrast, Full prompting grows rapidly ($\approx$ 3.6s@10, $\approx$ 4.7s@20, $\approx$ 7.7s@30) and becomes OOM beyond 30 turns. ICAE (append) is slower than ours at 10–20 turns and becomes OOM from 30 onward; ICAE (one-shot) is slower at 10 and becomes OOM already at 20. At 30 turns, ours is roughly $2.4\times$ faster than Full prompting. Most noticeably, *ours is the only approach that handles 428 turns*, reflecting the effectiveness of our novel retrieval-conditioned incremental compression. These results demonstrate our proposed C-DIC successfully scales to ever-growing, open-ended dialogue with strong memory and computational efficiencies. For the detailed latency components, see Table 11 in Appendix I. Note that OOM is setting-dependent: ICAE(append) scales primarily with the *number of turns*, whereas ICAE(one-shot) is constrained mainly by *total context length*. Consequently, their OOM thresholds can differ between the *all-sessions* latency evaluation here and the *per-session* results in Table 1.

### 4.6 ABLATIONS

We ablate three components of our system: *incremental compression* (IC), *retrieval-aware truncated BPTT* (R-TBPTT), and the *memory module* (retrieval + write-back) for context threading. As shown in Table 3, removing IC causes the greatest degradation. PPL rises from **9.356** to **25.527** and ROUGE-2 drops from **0.056** to **0.018**, indicating that turn-wise compression and revision are critical for preserving salient content. Disabling R-TBPTT also degrades supervision quality, confirming the benefit of backpropagating one hop along the actual retrieval path. Despite removing the memory-based context threading gives slightly better PPL, it yields markedly worse BLEU and recall (ROUGE scores), implying our proposed context threading greatly improves the long-

Table 3: **Ablation study on the REALTALK dataset.** We evaluate the contribution of incremental compression (IC), retrieval-aware truncated backpropagate through time (R-TBPTT) and memory by removing each component from the full model. All variants are evaluated at the final turn of each conversation under the *all-sessions* setting to assess long-term generation quality.

| Models | PPL↓ | BLEU↑ | R-L↑ | R-1↑ | R-2↑ |
|---|---|---|---|---|---|
| C-DIC | 9.356 | **0.069** | **0.173** | **0.213** | **0.056** |
| (–) Incremental Compression | 25.527 | 0.040 | 0.075 | 0.103 | 0.018 |
| (–) Retrieval-aware Truncated BPTT | 12.295 | 0.025 | 0.119 | 0.172 | 0.018 |
| (–) Memory-based Context Threading | **9.197** | 0.046 | 0.128 | 0.188 | 0.025 |

horizon coherence. In general, the full model (C-DIC) achieves state-of-the-art performance and demonstrates the necessity and complementary gains of all IC, R-TBPTT, and memory-based context threading.

## 5 CONCLUSION

In this paper, we present **Context-Driven Incremental Compression (C-DIC)**, a thread-aware dialogue memory for long conversations. It replaces full-context prompting with a lightweight *retrieve → revise → write-back* loop trained via retrieval-aware truncated BPTT, enabling cross-turn context sharing and revision without re-encoding the entire history. By modeling conversations as interleaved threads and retrieving only the compressed history relevant to the active context, C-DIC retains what matters and discards what does not, maintaining contextual fluency at low cost. Empirically, it remains stable where static compressors collapse under multi-turn rollout, outperforming truncation, summarization, and static latent baselines while reducing inference costs. It sustains nearly flat end-to-end latency (approximately 3∼3.5s) despite growing dialogue history, and is the only method demonstrated to handle up to 428 turns, underscoring the scalability of our incremental, retrieval-conditioned design. Ablations confirm that each component—incremental compression, retrieval-aware TBPTT, and memory-based context threading—contributes materially to the overall gains in coherence and faithfulness.

C-DIC has several limitations that we leave for future work. In this paper, we focus on long-term dialogue generation in open-domain chit-chat settings, and we do not evaluate on long-horizon, domain-specific tasks (e.g., medical advice, factual QA, coding assistants). Extending C-DIC to such domains remains an important direction. In addition, our datasets contain conversations of up to roughly 400 turns, which makes it challenging to conduct reliable human preference studies, since annotators would need to read and reason over very long histories. Designing scalable human (or LLM-as-judge) evaluation protocols tailored to such ultra-long dialogues is another key avenue for future work.

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

## A    KEY-VALUE CACHING

A standard engineering mitigation for inference is *key–value (KV) caching*, which stores the hidden states of past tokens layer-by-layer so that only the newest tokens are processed afresh (Tay et al., 2022). Caching indeed reduces *compute* for *unchanged* prefixes, but it introduces several limitations that are critical in an interactive setting. First, any user edit invalidates the cache from the edit point onward, which forces a full recomputation of attention. Second, KV caching trades FLOPs for memory: the cache footprint grows linearly with both dialogue length and layer count, quickly exhausting GPU memory when thousands of sessions run concurrently. Third, caching leaves the *attention distribution* unchanged, so the model still under-attends to mid-history tokens, a known positional-bias issue that harms long-range coherence.

## B    IMPLEMENTATION DETAILS

We implement our models with the `Llama2-Chat-7B` backbone by adapting the ICAE (Ge et al., 2024) checkpoint to the multi-turn dialogue setting. This provides a strong initialization for compression and allows us to focus on the effect of incremental, thread-aware memory. For practicality and to isolate the contribution of the memory mechanism, we freeze all weights of the base model and update only the LoRA-adapted compressor and the learnable compression tokens during training. All training and inference were conducted using an NVIDIA A100 80GB GPU. Fine-tuning required around 17 GPU hours on a single A100 GPU. Across all experiments employing the ICAE checkpoint, we use a compression token length of 128, a retrieval threshold $\tau = 0.8$, and cosine similarity with exponential decay (decay rate $\alpha = 0.05$. For training, we use a batch size of 1 while inference is performed with batch sizes of 8 and 1 for the MSC and REALTALK datasets, respectively. We finetune our model for 2 epochs, using AdamW with a learning rate of $2 \times 10^{-4}$.

## C    ALTERNATIVE MEMORY UPDATE STRATEGIES

We implemented two alternatives to the simple replacement-based write-back: (i) Exponential Moving Average (EMA) updates with decay factors $\beta \in \{0.3, 0.5, 0.7\}$, using $\beta \cdot$ old_memory $+ (1 - \beta) \cdot$ new_memory ,and (ii) a 2-layer gating network that learns to interpolate between the old and new memory states. As shown in Table 4, EMA brings at best marginal gains only on the R-1 metric

and often yields noticeably worse performance on other metrics, while the gated variant provides only small improvements on some ROUGE scores at the cost of additional complexity. Given this trade-off, we adopt the replacement policy as the simpler and more robust choice.

Table 4: **Comparison of memory update strategies on MSC-session 5.** Performance as a function of the selection threshold $\tau$. We report PPL↓, BLEU↑, ROUGE-L↑, ROUGE-1↑, and ROUGE-2↑.

| Strategy | PPL↓ | BLEU↑ | R-L↑ | R-1↑ | R-2↑ |
|---|---|---|---|---|---|
| Replacement | **8.427** | **0.030** | 0.160 | 0.206 | **0.040** |
| EMA ($\beta = 0.3$) | 8.442 | 0.027 | 0.157 | 0.208 | 0.035 |
| EMA ($\beta = 0.5$) | 8.836 | 0.027 | 0.155 | 0.203 | 0.036 |
| EMA ($\beta = 0.7$) | 9.929 | 0.021 | 0.145 | 0.186 | 0.030 |
| Gate | 8.503 | 0.026 | **0.162** | **0.209** | 0.037 |

## D  LIMITATIONS OF FULL BPTT AND FIXED-WINDOW TBPTT FOR DIALOGUE MEMORY

Under full BPTT, the gradient with respect to a memory slot $Z_s$ aggregates contributions from all future turns where that slot is actually consulted ($Z_s \in R_t$):

$$\frac{\partial L}{\partial Z_s} = \sum_{t=1}^{T} \mathbf{1}[Z_s \in R_t] \frac{\partial \ell_t}{\partial Z_s}. \tag{9}$$

However, implementing full BPTT requires keeping the computation graph for all $T$ turns in memory, so the activation cost grows linearly with dialogue length. For long conversations this becomes prohibitive in practice and quickly leads to out-of-memory (OOM) errors.

In standard fixed-window TBPTT with horizon $K$, at each turn $t$ all slots older than $K$ steps are detached from the computation graph before retrieval. Concretely, retrieval reads

$$\tilde{Z}_s = \begin{cases} \text{stopgrad}(Z_s), & s \le t - K, \\ Z_s, & s > t - K, \end{cases} \qquad R_t \subset \{\tilde{Z}_s\}_{s<t}. \tag{10}$$

By the chain rule, for any $s \le t - K$,

$$\frac{\partial \ell_t}{\partial Z_s} = \frac{\partial \ell_t}{\partial \tilde{Z}_s} \frac{\partial \tilde{Z}_s}{\partial Z_s} = \frac{\partial \ell_t}{\partial \tilde{Z}_s} \cdot 0 = 0, \tag{11}$$

even if $Z_s \in R_t$ (i.e., the slot is selected and used at turn $t$). The truncated gradient therefore becomes

$$\frac{\partial L}{\partial Z_s}\Big|_{\text{TBPTT}} = \sum_{t=1}^{T} \mathbf{1}[Z_s \in R_t]\, \mathbf{1}[t - s < K] \frac{\partial \ell_t}{\partial Z_s}, \tag{12}$$

so any selected memory state $Z_s$ that is retrieved only after it falls outside the $K$-step window receives no gradient signal from those distant uses.

## E  DATASET CHARACTERIZATION AND ANNOTATION RELIABILITY

We perform additional analysis to quantify (i) how often reference responses depend on distant context in MSC/REALTALK and (ii) how frequently target responses are generic, along with human verification of the LLM judge.

### E.1  DO MSC / REALTALK REQUIRE DISTANT CONTEXT?

**LLM-based annotation.**  We use GPT-4o to label whether a *candidate past utterance* contains **necessary or materially helpful** information for producing the **reference assistant response** to a target query. To avoid degraded judge reliability on very long prompts, we adopt a pairwise protocol: each instance consists of (i) one historical utterance and (ii) the final-turn context (latest user query

+ reference response), and the model outputs a binary label (`helpful` / `not helpful`).[1] We then aggregate utterance-level labels into **conversation-level** statistics (e.g., whether any supporting utterance occurs $\geq 10$ turns back).

**LLM-based annotation (genericity).**    Separately, we label each reference response as **generic** vs. **not-generic**.[2]. A response is *generic* if it can be plausibly reused across many different queries with minimal editing; otherwise it is *not-generic*.

**Sampling.**    We sample 500 and 320 conversations from MSC and REALTALK, respectively, restricting to dialogues with $\geq 11$ turns so that "$\geq 10$ turns back" is well-defined. These sample sizes provide stable estimation of conversation-level rates at reasonable cost (worst-case 95% margin $\approx \pm 4$–6 percentage points), consistent with cost-aware yet reliable LLM annotation practice (Pangakis et al., 2023).

**Results.**    As shown in Table 5, long-range dependencies are common in both datasets under these measures, while generic targets are rare. This suggests that strong performance on MSC/REALTALK is unlikely to be explained solely by short-range cues or templated responses.

Table 5: Dataset characterization via GPT-4o annotation. $n$ = sampled dialogues. Evid. $\geq 10$: fraction of *supporting utterances* occurring $\geq 10$ turns before the final response. Farthest $\geq 10$: fraction of dialogues whose most distant supporting utterance is $\geq 10$ turns back. Generic: fraction of target responses labeled generic.

| Dataset | n | Evid. $\geq 10$ (%) | Farthest $\geq 10$ (%) | Generic (%) |
|---|---|---|---|---|
| REALTALK | 320 | 66.94 | 40.31 | 6.25 |
| MSC | 500 | 44.92 | 70.80 | 2.00 |

### E.2    HUMAN VERIFICATION OF JUDGE RELIABILITY

To validate the GPT-4o labels, we run a human verification study with three annotators on 50 randomly sampled items per task and dataset, following the recommended LLM-as-a-judge verification setting in (Calderon et al., 2025). We provide the human verification guidelines and summarize the verification results below.

#### E.2.1    HUMAN VERIFICATION GUIDELINES

**A. Helpful-turn verification (utterance-level).**    You are shown: (1) the latest user query, (2) the final assistant reference response, (3) one past utterance from the same dialogue, and (4) the LLM label: `helpful` / `not helpful`.

- **helpful**: the past utterance contains information that is *necessary or clearly useful* to produce the final assistant response to the latest query (e.g., key facts, entities, constraints, preferences, or clarifications that the response depends on).

- **not helpful**: removing the past utterance would not materially change a reasonable final response (e.g., irrelevant details, small talk).

Mark **Correct = 1** if the LLM's `helpful`/`not helpful` label matches your judgment under the above definition; else **Correct = 0**.

---

[1]**Instruction 1 (utterance relevance).** You are given the latest user query and the assistant's response of a conversation along with an utterance from the past conversation. Your task is to determine if the utterance is helpful to generating the assistant's response to the latest user query. If it is helpful, respond with `helpful`; otherwise, respond with `not helpful`.

[2]**Instruction 2 (generic response).** You are given the latest user query and the assistant's final response of a conversation. Decide whether the assistant's final response is generic. A response is `generic` if it could be pasted into many different conversations/questions with minimal editing (e.g., greetings/farewells/small talk). Otherwise `not generic`. Output only generic or not generic.

**B. Generic-response verification (final-response-level).** You are shown: (1) the latest user query, (2) the final assistant reference response, and (3) the LLM label: `generic` / `not generic`.

- **`generic`**: the response could be pasted into many different conversations/queries with minimal edits (e.g., greetings/small talk, vague encouragement, "it depends" without specifics, generic steps not tailored to the query).

- **`not generic`**: the response provides concrete details, constraints, specific recommendations/decisions.

Mark **Correct = 1** if the LLM's `generic`/`not generic` label matches your judgment under the above definition; else **Correct = 0**.

### E.2.2 HUMAN VERIFICATION RESULTS

Table 6 shows the human verification results. GPT-4o labels match human judgments with high accuracy (about 90–96% accuracy) and strong inter-annotator consistency, supporting the reliability of our LLM-based labels for dataset-level characterization.

Table 6: Human verification of GPT-4o annotations (three annotators; 50 items per task and dataset). ACC is accuracy against the human majority vote. Agree is observed inter-annotator agreement. Fleiss' $\kappa$ measures agreement beyond chance.

| | MSC | | | REALTALK | | |
|---|---|---|---|---|---|---|
| **Task** | **ACC** (%) | **Agree** | **Fleiss'** $\kappa$ | **ACC** (%) | **Agree** | **Fleiss'** $\kappa$ |
| Helpful-turn label | 92.000 | 0.920 | 0.527 | 90.000 | 0.987 | 0.921 |
| Generic-response label | 95.918 | 0.973 | 0.652 | 96.000 | 0.973 | 0.653 |

## F ANALYSIS OF INCREMENTAL ICAE FAILURE MODES

The incremental ICAE variant fails for structural reasons (latent drift arising from repeatedly applying a one-shot objective) whereas ICAE (append) that simply accumulates compressed context and ICAE (one-shot) that re-encodes the full available context each turn work at short and medium lengths but run out of memory on very long conversations.

ICAE is trained with a one-shot compression objective: it learns to encode a contiguous context span into a latent in a single step. In the incremental variant, however, we repeatedly apply this compressor to its own compressed outputs as the dialogue progresses. This leads to latent drift and error compounding, because the model is never trained to use or update *already-compressed* contexts. Empirically, the response quality degrades rapidly across turns, as shown in Figure 1a.

By contrast, ICAE (append) and ICAE (one-shot) perform reasonably well at short and medium lengths, but eventually run out of memory on very long conversations as shown in Table 11 and Figure 3. In other words, ICAE (append) and ICAE (one-shot) are effective but not scalable, whereas the naive incremental application is scalable but unstable. This mismatch is exactly what motivates our design: C-DIC modifies the architecture and training scheme to support *incremental* use of compression while mitigating the catastrophic degradation observed in incremental ICAE.

## G MULTI-SEED ROBUSTNESS AND SIGNIFICANCE TESTS

To assess robustness to random initialization and stochastic training effects, we repeat all MSC and REALTALK experiments with three random seeds (42/43/44). We report mean±std across seeds for PPL, BLEU, and ROUGE-1/2/L. For REALTALK, results are reported in the *per-session* setting due to GPU memory limits of certain compression baselines under the full long-context setup (see Figure 3 and Table 11).

## G.1 Mean±std Across Seeds

Tables 7 and 8 summarize mean±std over three runs on MSC and REALTALK, respectively. Across three independent runs, our method exhibits low run-to-run variance (e.g., PPL std $\approx 0.04$ on both datasets) while consistently outperforming the strongest baseline ICAE(one-shot) across all reported metrics.

Table 7: MSC results (mean±std over seeds 42/43/44).

| Models | PPL↓ | BLEU↑ | R-L↑ | R-1↑ | R-2↑ |
|---|---|---|---|---|---|
| AutoCompressor | 9.109±0.273 | 0.014±0.002 | 0.121±0.002 | 0.145±0.003 | 0.021±0.001 |
| ICAE (incremental) | 561.702±347.397 | 0.007±0.001 | 0.063±0.006 | 0.075±0.008 | 0.005±0.001 |
| ICAE (one-shot) | 29.188±1.371 | 0.017±0.000 | 0.132±0.001 | 0.188±0.002 | 0.027±0.001 |
| **Ours** | **8.385±0.042** | **0.025±0.002** | **0.159±0.001** | **0.202±0.003** | **0.037±0.000** |

Table 8: REALTALK results (mean±std over seeds 42/43/44). Results are in the *per-session* setting.

| Models | PPL↓ | BLEU↑ | R-L↑ | R-1↑ | R-2↑ |
|---|---|---|---|---|---|
| AutoCompressor | 12.283±0.352 | 0.020±0.001 | 0.090±0.030 | 0.138±0.003 | 0.020±0.001 |
| ICAE (incremental) | 135.827±39.254 | 0.019±0.001 | 0.071±0.003 | 0.089±0.003 | 0.013±0.001 |
| ICAE (one-shot) | 25.115±3.388 | 0.025±0.001 | 0.113±0.005 | 0.159±0.006 | 0.024±0.002 |
| **Ours** | **9.764±0.043** | **0.034±0.001** | **0.136±0.002** | **0.177±0.002** | **0.032±0.001** |

## G.2 Seed-level Significance Tests

As additional evidence that gains are not driven by a favorable seed, we compute p-values using a paired t-test on the *seed-wise differences* between our method and ICAE(one-shot) (three paired observations).Table 9 reports the resulting p-values. For PPL, we apply the test on $\log(\text{PPL})$ to reflect the likelihood (average NLL) scale. Against ICAE(one-shot), improvements are statistically significant on both datasets (all $p < 0.05$).

Table 9: Paired t-test p-values for **Ours** vs. ICAE(one-shot) across three seeds (42/43/44).

| Dataset | $\log(\text{PPL})$ | BLEU | R-L | R-1 | R-2 |
|---|---|---|---|---|---|
| MSC | $2.8\times10^{-4}$ | 0.010 | $9.6\times10^{-5}$ | 0.003 | $5.6\times10^{-4}$ |
| REALTALK | 0.004 | 0.002 | 0.013 | 0.027 | 0.024 |

## H Closed-loop evaluation

This section reports additional results under *closed-loop* generation, where the assistant's past turns in the context are replaced by the model's own previously generated responses (user turns are kept fixed from the dataset). This evaluation explicitly stress-tests error accumulation under self-conditioned history.

Table 10 evaluates all methods on REALTALK in the *per-session* setting. We use per-session to keep methods comparable and largely runnable: several compression baselines exceed GPU memory under the full long-context (*all-sessions*) configuration, and ICAE(append) is OOM even in per-session (reported explicitly).

As shown in Table 10, our method achieves the best overall performance among runnable baselines across all reported metrics. Compared with the teacher-forcing results in Table 1, scores decrease slightly, which is expected under closed-loop generation due to compounding errors and occasional user–assistant misalignment on offline corpora. Nevertheless, the degradation for our method is modest, providing evidence that C-DIC remains stable when conditioning on its own generations.

Table 10: Closed-loop results on REALTALK (per-session).

| Models | PPL↓ | BLEU↑ | R-L↑ | R-1↑ | R-2↑ |
|---|---|---|---|---|---|
| Full prompting | 29.666 | 0.020 | 0.106 | 0.150 | 0.017 |
| Truncation | 28.174 | 0.021 | 0.109 | 0.156 | 0.019 |
| Summarization | 27.977 | 0.022 | 0.110 | 0.162 | 0.021 |
| In-Session RAG | 26.789 | 0.020 | 0.103 | 0.149 | 0.015 |
| AutoCompressor | 13.111 | 0.020 | 0.111 | 0.144 | 0.021 |
| ICAE (incremental) | 124.024 | 0.020 | 0.068 | 0.088 | 0.012 |
| ICAE (one-shot) | 17.364 | 0.024 | 0.109 | 0.152 | 0.023 |
| ICAE (append) | | | *Out of memory* | | |
| **Ours** | **9.754** | **0.034** | **0.133** | **0.173** | **0.031** |

## I    DETAILED LATENCY COMPONENTS

Table 11 decomposes end-to-end latency on REALTALK-*all sessions* as the maximum number of preserved turns increases ($\{10, 20, 30, 40, 428\}$). We report *Comp. Time* (time spent on context preparation such as compression/selection) and *Gen. Time* (model runtime to produce the response); *Total Time* is their sum. All values are in seconds. *Out of memory* indicates a method failed at that context length.

Table 11: **Latency components depending on the max number of turns.** Compression (**Comp. Time**) and generation (**Gen. Time**) times for ICAE (one-shot), ICAE (append), and C-DIC (Ours) on REALTALK at maximum turns $\{10, 20, 30, 40, 428\}$. The total latency equals **Comp. Time + Gen. Time** All values are in seconds. *Out of memory* indicates the baseline failed to run at that maximum turn due to the context length.

| Models | Max # of Turns | Comp. Time | Gen. Time | Total Time |
|---|---|---|---|---|
| Full prompting | | 0.00 | 3.53 | 3.53 |
| ICAE (one-shot) | 10 | 0.26 | 3.97 | 4.24 |
| ICAE (append) | | 0.11 | 3.53 | 3.64 |
| **Ours** | | **0.05** | **3.16** | **3.21** |
| Full prompting | | 0.00 | 4.81 | 4.81 |
| ICAE (one-shot) | 20 | | *Out of memory* | |
| ICAE (append) | | 0.27 | 3.97 | 4.24 |
| **Ours** | | **0.05** | **3.28** | **3.33** |
| Full prompting | | 0.00 | 7.66 | 7.66 |
| ICAE (one-shot) | 30 | | *Out of memory* | |
| ICAE (append) | | | *Out of memory* | |
| **Ours** | | **0.05** | **3.21** | **3.26** |
| Full prompting | | | *Out of memory* | |
| ICAE (one-shot) | 40 | | *Out of memory* | |
| ICAE (append) | | | *Out of memory* | |
| **Ours** | | **0.05** | **3.34** | **3.40** |
| Full prompting | | | *Out of memory* | |
| ICAE (one-shot) | 428 | | *Out of memory* | |
| ICAE (append) | | | *Out of memory* | |
| **Ours** | | **0.06** | **3.30** | **3.36** |

## J    ABLATION ON RETRIEVAL THRESHOLD

In C-DIC, the retrieval threshold $\tau$ is not directly optimized but interacts with **learned** representations: during training, the encoder and memory updater adapt such that cosine similarities between the query and relevant memory states evolve in a way that is compatible with a fixed decision boundary $\tau$. As shown in Table 12, we empirically find that performance is stable for a broad range of values (roughly 0.2–0.8); only very low thresholds (which make almost all states "similar") or very high thresholds (which make almost no states "similar") lead to noticeable degradation, because the model either over-updates or under-utilizes memory.

Table 12: **Performance as a function of the selection threshold $\tau$.** Performance as a function of the selection threshold $\tau$. We report PPL, BLEU, ROUGE-L, ROUGE-1, and ROUGE-2, evaluated on MSC-session 5 at the final dialogue turn to assess long-term generation quality.

| Threshold | PPL↓ | BLEU↑ | R-L↑ | R-1↑ | R-2↑ |
|---|---|---|---|---|---|
| 0.1 | 9.593 | 0.022 | 0.150 | 0.193 | 0.033 |
| 0.2 | 8.351 | 0.025 | 0.155 | 0.203 | 0.034 |
| 0.3 | 8.334 | 0.027 | 0.162 | 0.212 | 0.041 |
| 0.4 | 8.346 | 0.028 | 0.159 | 0.204 | 0.039 |
| 0.5 | 8.362 | 0.026 | 0.156 | 0.203 | 0.036 |
| 0.6 | 8.325 | 0.028 | 0.157 | 0.208 | 0.037 |
| 0.7 | 8.383 | 0.027 | 0.156 | 0.207 | 0.036 |
| 0.8 | 8.427 | 0.030 | 0.160 | 0.206 | 0.040 |
| 0.9 | 12.202 | 0.022 | 0.139 | 0.190 | 0.027 |

## K    EFFECT OF COMPRESSION TOKEN LENGTH

This work studies *incremental* compression for multi-turn dialogue by extending ICAE (Ge et al., 2024), rather than re-optimizing the underlying compressor for static-document compression. Accordingly, unless otherwise stated, our main experiments use the publicly released ICAE checkpoint, which provides a single compression length (128 tokens).

To examine the impact of compression length in the multi-turn setting, we train ICAE-style compressors with 64, 128, and 256 tokens on MSC (Xu et al., 2022) using the same ICAE training objectives (one-shot continuation and auto-encoding). We then fine-tune the dialogue model with our proposed method while varying only the compression token budget. Table 13 reports results on MSC under the 5-session evaluation setting.

Table 13: **Comparison of compression token lengths on MSC-session 5.** Performance as a function of the selection threshold $\tau$. We report PPL↓, BLEU↑, ROUGE-L↑, ROUGE-1↑, and ROUGE-2↑.

| Comp. Token Length | PPL↓ | BLEU↑ | R-L↑ | R-1↑ | R-2↑ |
|---|---|---|---|---|---|
| 64 | 8.604 | 0.023 | 0.157 | 0.201 | 0.036 |
| 128 | 8.582 | 0.023 | 0.155 | 0.200 | 0.034 |
| 256 | 8.646 | 0.022 | 0.160 | 0.205 | 0.037 |

These results indicate that performance is relatively stable across 64–256 tokens, with differences in PPL and generation metrics being modest and without a clear monotonic trend. This suggests that C-DIC is not overly sensitive to the exact compression capacity within this range.

## L    LONGMEMEVAL: LONG-CONTEXT QA EVALUATION

This appendix reports an additional evaluation on LONGMEMEVAL$_S$, a benchmark designed to assess long-term memory and question answering in chat assistants. We evaluate in a **zero-shot** setting using the same LLM backbone across all methods (`Llama-2-Chat-7B`). We compare (i) full prompting and (ii) latent-compression baselines (ICAE variants), against (iii) our CDIC-based approach. Following the LONGMEMEVAL protocol, we use **GPT-4o** as the automatic judge to determine answer correctness and report Accuracy.

Table 14 shows that CDIC yields the best accuracy among the compared methods, improving over full prompting while using substantially fewer input tokens. These results provide evidence that CDIC improves QA performance under long-context settings, complementing our dialogue-generation results on MSC and REALTALK.

## M    REALTALK: TWO-SESSION EVALUATION

The REALTALK (Lee et al., 2025) dataset contains substantially longer multi-session dialogues than MSC (Xu et al., 2022). In Table 1, we therefore report results *per session* to avoid out of

Table 14: **LongMemEval results (zero-shot).** All methods use the same backbone (LLaMA2-7B). Accuracy is computed by a GPT-4o judge following the LONGMEMEVAL evaluation protocol.

| Models | Accuracy↑ |
|---|---|
| Full prompting | 0.086 |
| ICAE (incremental) | 0.010 |
| ICAE (one-shot) | 0.004 |
| **Ours** | **0.116** |

memory issue. For completeness, Table 15 reports performance on the subset of REALTALK conversations with **two sessions**. We follow the same evaluation protocol and hyperparameters as in the main results. Note that our model shows consistent performance without the session limit in Figure 4.

Table 15: **REALTALK two-session results.** Test performance on conversations with up to two sessions. Lower is better for PPL; higher is better for BLEU/ROUGE.

| Models | PPL↓ | BLEU↑ | R-L↑ | R-1↑ | R-2↑ |
|---|---|---|---|---|---|
| Full prompting | 27.225 | 0.022 | 0.109 | 0.159 | 0.020 |
| Truncation | 21.865 | 0.026 | 0.109 | 0.174 | 0.028 |
| Summarization | 26.120 | 0.023 | 0.118 | 0.172 | 0.025 |
| In-Session RAG | 27.435 | 0.019 | 0.100 | 0.145 | 0.014 |
| AutoCompressor | 11.150 | 0.020 | 0.112 | 0.146 | 0.022 |
| ICAE (incremental) | 218.103 | 0.017 | 0.059 | 0.073 | 0.007 |
| ICAE (one-shot) | | *Out of memory* | | | |
| ICAE (append) | | *Out of memory* | | | |
| **Ours** | **9.870** | **0.034** | **0.132** | **0.175** | **0.029** |

## N    ADDITIONAL RESULTS ON MULTI-SESSION CHAT (MSC)

Table 16 reports a detailed breakdown of model quality setting across different session lengths (2–5) as well as the aggregate over all sessions. We evaluate generation with perplexity (PPL; lower is better) and text-overlap metrics (BLEU, ROUGE-L/1/2; higher is better).

## O    QUALITATIVE EXAMPLES

We present one of the qualitative examples demonstrating C-DIC's effectiveness in context coherence in a long dialogue setting in Figure 7 and 8.

Table 16: Comparison across MSC sessions. Lower is better for PPL; higher is better for others.

| Models | Session | PPL ↓ | BLEU ↑ | R-L ↑ | R-1 ↑ | R-2 ↑ |
|---|---|---|---|---|---|---|
| Full prompting | AVG | 41.245 | 0.008 | 0.110 | 0.157 | 0.015 |
| Truncation | | 30.890 | 0.012 | 0.128 | 0.184 | 0.024 |
| Summarization | | 41.849 | 0.013 | 0.128 | 0.172 | 0.024 |
| In-Session RAG | | 35.530 | 0.008 | 0.110 | 0.148 | 0.014 |
| AutoCompressor | | 9.285 | 0.012 | 0.121 | 0.174 | 0.021 |
| ICAE (incremental) | | 513.774 | 0.006 | 0.057 | 0.069 | 0.005 |
| ICAE (one-shot) | | 27.656 | 0.017 | 0.133 | 0.190 | 0.027 |
| **Ours** | | **8.431** | **0.023** | **0.160** | **0.205** | **0.037** |
| Full prompting | 5 | 40.801 | 0.012 | 0.113 | 0.165 | 0.016 |
| Truncation | | 26.252 | 0.014 | 0.136 | 0.198 | 0.028 |
| Summarization | | 38.9759 | 0.0148 | 0.129 | 0.1881 | 0.024 |
| In-Session RAG | | 33.931 | 0.009 | 0.112 | 0.163 | 0.014 |
| AutoCompressor | | 9.364 | 0.012 | 0.123 | 0.151 | 0.020 |
| ICAE (incremental) | | 442.062 | 0.006 | 0.061 | 0.074 | 0.004 |
| ICAE (one-shot) | | 30.312 | 0.016 | 0.134 | 0.196 | 0.027 |
| **Ours** | | **8.553** | **0.024** | **0.160** | **0.211** | **0.038** |
| Full prompting | 4 | 40.621 | 0.009 | 0.109 | 0.157 | 0.014 |
| Truncation | | 26.427 | 0.013 | 0.133 | 0.188 | 0.024 |
| Summarization | | 40.162 | 0.012 | 0.130 | 0.190 | 0.024 |
| In-Session RAG | | 33.318 | 0.008 | 0.110 | 0.161 | 0.014 |
| AutoCompressor | | 9.222 | 0.012 | 0.129 | 0.150 | 0.020 |
| ICAE (incremental) | | 454.005 | 0.006 | 0.061 | 0.074 | 0.004 |
| ICAE (one-shot) | | 29.261 | 0.016 | 0.133 | 0.194 | 0.028 |
| **Ours** | | **8.418** | **0.022** | **0.158** | **0.205** | **0.037** |
| Full prompting | 3 | 40.221 | 0.009 | 0.110 | 0.157 | 0.014 |
| Trunc.-5 | | 27.678 | 0.014 | 0.127 | 0.192 | 0.020 |
| Summarization | | 42.804 | 0.012 | 0.127 | 0.184 | 0.024 |
| In-Session RAG | | 35.751 | 0.008 | 0.110 | 0.158 | 0.018 |
| AutoCompressor | | 9.222 | 0.012 | 0.123 | 0.150 | 0.020 |
| ICAE (incremental) | | 505.368 | 0.006 | 0.058 | 0.070 | 0.004 |
| ICAE (one-shot) | | 26.971 | 0.018 | 0.137 | 0.195 | 0.029 |
| **Ours** | | **8.350** | **0.023** | **0.162** | **0.206** | **0.037** |
| Full prompting | 2 | 43.323 | 0.003 | 0.109 | 0.153 | 0.013 |
| Truncation | | 29.414 | 0.013 | 0.133 | 0.188 | 0.023 |
| Summarization | | 45.456 | 0.012 | 0.126 | 0.183 | 0.024 |
| In-Session RAG | | 39.120 | 0.008 | 0.108 | 0.148 | 0.017 |
| AutoCompressor | | 9.333 | 0.011 | 0.110 | 0.149 | 0.017 |
| ICAE (incremental) | | 653.668 | 0.006 | 0.058 | 0.070 | 0.004 |
| ICAE (one-shot) | | 24.081 | 0.018 | 0.130 | 0.174 | 0.025 |
| **Ours** | | **8.404** | **0.022** | **0.159** | **0.198** | **0.037** |

Figure 5: **Qualitative example #1 (LongMemEval).** Multi-session dialogue in chronological order (previous sessions → current session).

---

**Dialogue Snippet**

**Session 1∼11**

*. . . (56 turns) . . .*

**Session 12**

*. . . (1 turn) . . .*

**Turn #58**

**S1:** I'm also thinking of exploring other platforms like Instagram and Twitter to promote my writing services. Do you have any tips on how to get started with those platforms, especially since I don't have much experience with them? By the way, speaking of first purchases, I remember getting a **10% discount** on my first purchase from that new clothing brand last month, which was nice.

**S2:** Great mention about the discount! Who doesn't love a good deal, right? Now, let's dive into getting started with Instagram and Twitter to promote your writing services. Both platforms have unique features and audiences, so I'll provide you with some tips to help you get started: **Instagram:** 1. **Visual content is key**: Instagram is all about visuals, so focus on creating high-quality images, graphics, or videos that showcase your writing skills, books, or writing process. 2. **Use hashtags strategically**: Research and use relevant hashtags in your captions to increase visibility. You can use a mix of niche-specific hashtags (e.g., #writingcommunity, #freelancewriting) and broader hashtags (e.g., #writerslife, #booklovers). 3. **Create an Instagram Stories strategy**: Share behind-the-scenes content, sneak peeks, or Q&A sessions to engage with your audience. You can also use the "swipe-up" feature to drive traffic to your website or other social media profiles. 4. **Leverage Instagram's features**: Use IGTV for longer-form content, Reels for short videos, and IG Live for real-time engagement. 5. **Collaborate with other writers or influencers**: Partner with fellow writers or influencers in your niche to reach new audiences and create content that appeals to a broader range of people. 6. **Post consistently**: Aim for at least 3-5 posts per week, but make sure you don't sacrifice quality for quantity. 7. **Engage with your audience**: Respond to comments, answer questions, and use Instagram's features like polls and quizzes to encourage interaction. **Twitter:** 1. **Keep it concise**: Twitter is all about short, bite-sized content. Keep your tweets concise, informative, and engaging. 2. **Use hashtags**: Research and use relevant hashtags in your tweets to increase visibility. You can use a mix of niche-specific hashtags (e.g., #writingtips, #freelancewriting) and broader hashtags (e.g., #writerslife, #books). 3. **Tweet regularly**: Aim for at least 5-10 tweets per week, but make sure you don't overwhelm your audience. 4. **Engage with others**: Respond to tweets, answer questions, and participate in Twitter Chats to build relationships with other writers, influencers, and potential clients. 5. **Share a mix of content**: Share your own content, retweet relevant articles or tweets, and participate in Twitter polls to keep your audience engaged. 6. **Use Twitter Ads**: Consider running targeted ads to increase your reach and engagement, especially if you're just starting out. 7. **Optimize your profile**: Make sure your Twitter profile is complete, including a profile picture, header image, and bio that clearly states your writing services and expertise. **Additional Tips:** 1. **Cross-promote**: Share your Instagram and Twitter content on your other social media platforms, like Facebook and LinkedIn, to reach a broader audience. 2. **Be authentic**: Be yourself, and let your personality shine through on both platforms. This will help you build a loyal following and attract potential clients who resonate with your voice and style. 3. **Monitor and adjust**: Keep an eye on your analytics and adjust your strategy based on what's working and what's not. Remember, building a presence on Instagram and Twitter takes time and effort. Focus on creating high-quality content, engaging with your audience, and promoting your writing services consistently. Good luck!

*. . . (4 turn) . . .*

**Session 13∼48**

*. . . (187 turns) . . .*

---

**Question:** What was the discount I got on my first purchase from the new clothing brand?

**Full prompting:** "I'm just an AI, I don't have access to personal information or purchase history, . . . "

**ICAE (incremental):** "I am really excited to see what kind of discount you got given on your first purchase from the new clothing brand!"

**ICAE (one-shot):** "southeast asian cultures: southeast asian cultures have a rich history of textile and print design, . . . "

**Ours:** "You got **10 %** off your first purchase!"
**Ground-truth:** "10 %"

Figure 6: **Qualitative example #2 (LongMemEval).** Multi-session dialogue in chronological order (previous sessions → current session).

---

**Dialogue Snippet**

---

**Session 1∼4**

... *(14 turns)* ...

**Session 5**

... *(1 turn)* ...

**Turn #16**

**S1:** I love these ideas! I'm definitely going to consider the Global Street Food theme. By the way, I've also had a great experience with a BBQ theme, like ==the one we had at Mike's place two weeks ago==, where we watched a football game together. Do you have any BBQ recipe ideas that would fit well with the Global Street Food theme?

**S2:** A BBQ theme can be a fantastic way to incorporate grilled and smoky flavors into your Global Street Food potluck. Here are some BBQ-inspired recipes ... *(continued)*

**S1:** That's great! I think I'll have a mix of grilled and non-grilled dishes to cater to different tastes. By the way, I've also had experience with dinner parties that are more low-key, like ==the ones we had at Alex's place yesterday==, where we had a potluck and tried out different cuisines from around the world, and ==also at Mike's place==, where we had a BBQ and watched a football game together. Do you have any tips on how to make the party more interactive and engaging for guests?

**S2:** You're referring to creating a more immersive and engaging experience for your guests. ... *(continued)*

... *(2 turns)* ...

**Session 6∼8**

... *(8 turns)* ...

**Session 9**

**Turn #28**

**S1:** I'm looking for some Italian recipe ideas for a dinner party I'm hosting soon. ==I attended a lovely Italian feast at Sarah's place last week==, and it inspired me to try out some new dishes. Do you have any recommendations for a beginner like me?

**S2:** Buon appetito! I'm thrilled to hear that Sarah's Italian feast inspired you to host your own dinner party. Italian cuisine is incredibly diverse and rich, so I'd be happy to help you with some recipe ideas perfect for a beginner. Here are some classic and delicious Italian dishes ... *(continued)*

**S1:** That's a great list of recipes! I think I'll start with the Spaghetti Aglio e Olio and Chicken Parmesan. I've always loved the combination of garlic and olive oil. By the way, speaking of dinner parties, ==I've had a lovely experience at Sarah's place recently==, where we played board games until late into the night after the Italian feast. Do you have any recommendations for board games that would be a good fit for a dinner party?

**S2:** Spaghetti Aglio e Olio and Chicken Parmesan are both delicious and crowd-pleasing options. Now, about board games for a dinner party ... *(continued)*

... *(4 turns)* ...

**Session 10∼44**

... *(190 turns)* ...

---

**Question:** How many dinner parties have I attended in the past month?

---

**Full prompting:** "I'm just an AI, I don't have access to your personal information or calendar, so I'm unable to accurately answer your question. However, I can suggest that you check your calendar or contact the hosts of any dinner parties you may have attended in the past month ..."

**ICAE (incremental):** "The number of the month? Is that 30! And if you're a math teacher you'll have 30 students!"

**ICAE (one-shot):** " Reduce, Reuse, Recycle: Implement the 3Rs in your daily life by reducing your consumption, ..."

**Ours:** "You've attended **3 dinner parties** in the past month."

**Ground-truth:** "Three"

Figure 7: **Qualitative example (MSC - session 5).** Multi-session dialogue in chronological order (previous sessions → current session).

**Dialogue (prev. sessions → current session)**

**Session 1**
S1: Hi, do you like movies? I really do.
S2: Omgosh, I love movies!
S1: Nice, we have something in common. How do you feel about pizza?
S2: I love pizza. But I love cooking healthy mostly.
S1: Oh, I try, but can't resist pizza and good burgers sometimes...
S2: Lol just because I cook healthy doesnt mean I do not splurge haha.
S1: Good to hear. What do you do? I only work part time now.
S2: I am a mom so I mainly watch my kiddos all day at the moment lol.
S1: That's a busy job. Mine is at a warehouse, that's a busy place, too.
S2: Oh I am sure that is alot of work!
S1: Yeah, that's why I can only handle part time. I need time for cruising...lol.
S2: I love cruising. I am living in california right now, great place to cruise.

**Session 2**
S1: My kids really wanted pizza so I got to have a bit of a cheat day.
S2: Oh I love pizza too! What's your favourite toppings?
S1: Anything with meat! . . . And extra onions . . . How about you?
S2: I love meat pizzas . . . chicken, ham, sweetcorn, pineapple.
S1: I LOVE stuffed crust, but I order without to watch my weight.
S2: Same here . . . guilty pleasure is takeaway with a movie.
S1: What kinds of movies do you watch together?
S2: Disney; the kids' favorite is Moana.
S1: I prefer comedy, can't wait to stop seeing Frozen!
S2: We switched to Moana. I'd love to watch horror.
S1: I had nightmares from Gremlins and Ghoulies as a kid.
S2: Jaws did that to me!

**Session 3**
S1: Great trailer for a horror movie—want to watch it with me?
S2: Yes! And let's sneak in a pepperoni, sausage, meatball pizza.
S1: No kids, grown-up movie and pizza—perfect!
S2: Add cold beer; need me to bring anything?
S1: Chocolate bars. Any movie prefs?
S2: Avoid slasher if possible.
S1: How about "A Quiet Place 2"?
S2: Sounds good. Then a comedy like *Horrible Bosses*.
S1: Maybe some healthy snacks too.
S2: Thumbs down—let's do cotton candy and bourbon!
S1: I'll bring Woodford Reserve.
S2: I have Blantons from a Buffalo Trace visit!

1242
1243
1244
1245
1246
1247
1248
1249
1250
1251
1252

Figure 8: **Qualitative example (continued).** Continuation of the dialogue and model outputs to the final question.

---

**Dialogue (continued)**

---

**Session 4**
S2: Was the movie scary enough? Recovered from bourbon?
S1: Still having nightmares; could've had more bourbon.
S2: There's never enough. **I'm recovering from injuring myself with pizza!**
S1: Oh no—did you cut yourself slicing it?
S2: Kids distracted me; missed a bone in the chicken and sliced my throat when I swallowed.
S1: You're joking?! Will that put you off chicken?
S2: No, I'll still put chicken on pizza, but my voice sounds like Freddy Kruger.
S1: LOL. Entertaining bedtime stories for the kids—NOT!
S2: They're scared but well behaved—it was almost worth it.
S1: Now that you know scary voices help, you could use it.
S2: I recommend it—without the injury. Still pineapple on your pizza?
S1: I'll be fine with chicken; not a pineapple-on-pizza person—extra cheese!
**Current session**
S1: Are your kids on any new movie kick recently?
S2: They've been really into Frozen lately.
S1: Same here—know every song by heart!
S2: Maybe we can all watch it together.
S1: Disney musical marathon—my fav is *Sound of Music*. Yours?
S2: I like *La La Land*.
S1: Haven't seen it. Is it a kid movie?
S2: It isn't—romantic musical with Leonardo DiCaprio and Emma Stone.
S1: Sounds good! I love Leonardo.
S2: Perfect movie for you; he's a surprisingly good singer.

---

**User (S1):** *By the way, which food caused my injury during cooking?*

---

**Full prompting:** "I'm just an AI, I don't have access to personal information . . ."

**Truncation:** "Oh, my goodness! . . . I'm afraid I can't reveal the identity of the food . . ."

**Summarization:** "I'm afraid I can't reveal the identity of the culinary criminal . . ."

**In-Session RAG:** "I apologize, but I'm just an AI and do not have access . . ."

**AutoCompressor:** "I think it was a pineapple."

**ICAE (incremental):** degenerates into repetition ("It . . . It . . .").

**ICAE (one-shot):** "I'm just an AI . . . be mindful and cautious when cooking . . ."

**ICAE (append):** *OOM*.

**Ours:** "Oh, sorry. It was the **pizza**. You slipped on cheeze."

---
