# OpenReview forum: "Finding the Thread: Context-Driven Incremental Compression for Multi-Turn Dialogue"
_ICLR.cc/2026/Conference — Submitted to ICLR 2026_

### Official Review · Reviewer_QNTu · 2025-10-24

**Soundness:** 3
**Presentation:** 3
**Contribution:** 2
**Rating:** 6
**Confidence:** 4

**Summary:**

The paper introduces Context-Driven Incremental Compression (CDIC), a context compression approach. It maintains  a compressed state at each turn using a memory module performing three basic operations: retrieve, revise, and write-back. The approach selectively retrieve the context relevant to the current turn instead  of re-encoding the entire history. The module is trained with truncated BPTT. It outperforms existing baselines such as truncation, summarization, and static compression.

**Strengths:**

- CDIC is a lightweight approach that does not required updating the parameters of the base model.
- There is an ablation that emphasizes the contribution of each component of the approach.
- CDIC is compared with diverse baselines.. It outperforms these existing baselines and generalizes out of distribution.

**Weaknesses:**

- Unlike some of the baselines, CDIC requires an additional fine-tuning
- The main problem that the paper aims to address is processing long-context input. However, the proposed approach is not evaluated on any recent long-context benchmarks (for example LoCoMo, MemoryBank, DuLeMon, PerLTQA, LongMemEval, MemBench ...). It is not known how models augmented with CDIC behave on very long contexts.
- The paper only evaluates using perplexity or metrics based on content matching. It i unclear whether CDIC allows the model to answer questions with exact answers (facts, math.coding, question answering).

**Questions:**

- How does the method perform on long-context benchmark compared to existing baselines?
- Does the model generalize on datasets from other domains (for example factual QA, math, coding) or is it required to perform fine-tuning on all these domains?
- Does CDIC affect the other capabilities of the model? Is ther catastrophic forgetting?
- Does R-TBPTT allow to train on very long inputs? How does the required memory increase as a function of input length?

---

> ### Author Response · Authors · 2025-11-21
> **Response to Reviewer QNTu**
>
> Thank you for your thoughtful review and for highlighting the strengths of our work. We sincerely appreciate your recognition that C-DIC is a lightweight framework that improves long-context handling *without* updating the base model’s parameters.
>
> **Q1: Unlike some of the baselines, CDIC requires an additional fine-tuning.**
>
> **A1:** We acknowledge that C-DIC involves an extra fine-tuning stage, but emphasize that:
> - **Only the memory module is updated, not the base LM.**
> We freeze the underlying LLM and fine-tune only the compressor / memory components, which are a small fraction of the total parameters. This keeps the computational cost and engineering burden much lower than full model fine-tuning.
> - **Incremental compression genuinely requires adaptation.**
> Static baselines (e.g., naive truncation or one-shot ICAE) were not designed for repeatedly compressed multi-turn dialogue; simply applying them “as is” leads to the failure modes we report. Our goal is precisely to study learnable, incremental memory, which necessarily involves training the memory module to operate in this setting.
> - **Training cost is one-off; efficiency gains are permanent.**
> The additional fine-tuning is a single, offline cost. Once trained, C-DIC can be paired with the frozen base LM for many deployments, while providing substantial inference-time benefits (constant latency with long histories and better performance than static compressors).
>
> **Q2: Missing evaluation on recent long-context benchmark.**
>
> **A2**: Our focus in this work is on **multi-turn dialogue** with incremental, turn-by-turn compression, so we prioritized MSC and REALTALK, which provide naturally long conversational histories and allow us to stress-test C-DIC over hundreds of turns. In particular, REALTALK contains some of the longest dialogues among mentioned benchmark datasets (average 447.1 turns), making it highly suitable for evaluating long-horizon conversational memory.
>
> However, we agree that more detailed evaluation of long-context tracking would further contextualize our approach, and we are currently working on this. In addition, we are conducting a dataset analysis that quantifies long-range context dependencies in MSC and REALTALK, to more clearly demonstrate that long-context tracking is indeed required for response generation in our setting.
>
> **Q3: Does the model generalize on datasets from other domains (for example factual QA, math, coding) or is it required to perform fine-tuning on all these domains?**
>
> A3: We are currently running experiments on long-context QA to probe how well C-DIC transfers beyond chit-chat. We will include a brief summary of these results in the revised version.
>
> **Q3: Is there catastrophic forgetting?**
>
>  **A3**: Some degree of forgetting is inevitable in long-context setting, and C-DIC is not an exception. However, our results indicate that C-DIC substantially mitigates catastrophic forgetting compared to both non-compression and static-compression baselines:
>
> - Compared to **no compression** (full prompting up to the context limit), C-DIC achieves **better or comparable generation quality**
> while maintaining constant latency and avoiding out-of-memory failures on very long dialogues.
>
> - Compared to **incremental ICAE** and other static compression schemes, C-DIC avoids the catastrophic degradation we observe in those models (e.g., extremely high PPL when repeatedly compressing already-compressed states).
>
> This suggests that, while some information loss still occurs, C-DIC’s thread-aware, revisable memory makes forgetting **much more controlled**, and in practice it yields *better* answers than both non-compression baselines under long-context constraints.
>
> **Q4: Does R-TBPTT allow to train on very long inputs? How does the required memory increases as a function of input length?**
>
>  **A3:** Yes. R-TBPTT is designed precisely so we can train on very long dialogues (hundreds of turns) without the activation memory blowing up.
>
> **During training, we never keep a full computation graph over the entire dialogue**. The activation memory we need is dominated by:
>
> 1. the current forward pass of the generator at turn t, and
>
> 2. the computation for updating the retrieved memory slot in reverse time.
>
> As a result, the required GPU memory for backpropagation is effectively **independent of the total number of dialogue turns** and does **not** grow with input length; it is bounded by the per-turn computation and the one-hop update, rather than by the full dialogue length.

---

> ### Author Response · Authors · 2025-12-03
>
> **Q2 (Missing long-context benchmarks) & Q3 (Lack of exact-answer evaluation).**
>
> We thank the reviewer for raising two related concerns:
>
> **(i)** our main experiments do not include several *recent* long-context benchmarks, leaving behavior under *very long* contexts unclear; and
>
> **(ii)** our primary metrics (PPL/BLEU/ROUGE) are based on content matching with reference and thus do not directly measure **exact-answer correctness** (*e.g.*, QA).
>
> **First, our current evaluation setup already stresses very long dialogue context**
>
> REALTALK is a real-world, multi-session WhatsApp-style corpus with very long conversations (avg. **21.9 sessions** and **447** **turns** per conversation). Among commonly used long-context dialogue datasets, REALTALK is **one of the longest in turns** and therefore directly stress-tests scalability in realistic multi-session conversations.
>
> **The reference-based metrics do implicitly reflect the "context tracking" ability as our benchmarks majorly consist of "specific" target responses, penalizing "generic" responses.**
>
> To validate that evaluations on MSC & REALTALK reflect long-range context tracking (rather than being solvable by short-range cues or templated responses), we present a dataset analysis that quantifies: proportions of **(i)** target responses relying on distant supporting turns, and **(ii)** target responses being generic.
>
> | Dataset | Sampled dialogues | Evidence from ≥10 turns (%) | Dialogues w/ farthest ≥10 (%) | Generic resp. (%) |
> | --- | :---: | :---: | :---: | :---: |
> | REALTALK | 320 | 66.94 | 40.31 | 6.25 |
> | MSC | 500 | 44.92 | 70.80 | 2.00 |
>
> “Evidence from ≥10 turns (%)” measures the proportion of *relevant turns* at ≥10 earlier turns to all relevant turns for response generation, while “Dialogues w/ farthest support ≥10 turns (%)” measures the ratio of dialogues having their relevant turns at ≥10 earlier turns to all dialogue samples. These results show that **long-range dependencies are common** and that **generic target responses are rare**, indicating that strong overlap-based scores (PPL, BLUE, ROUGE scores) reflect the long context tracking capability. Full sampling details and human verification results are provided in **Appendix E**.
>
> **Additional exact-answer long-context evaluation: LongMemEval (QA)**
>
> To directly address **both** benchmark coverage (Q2) and exact-answer evaluation (Q3), we additionally perform **zero-shot** evaluations on **LongMemEval** [1], which is designed to test long-term question answering in chat assistants. We compare (i) Full prompting and (ii) latent-compression baseline models (ICAE variants), against (iii) our C-DIC-based approach. Following the protocol of LongMemEval, we use **GPT-4o** as the automatic judge to measure the answer correctness.
>
> | Model | Accuracy ↑ |
> | --- | :---: |
> | Full prompting (base model) | 0.086 |
> | ICAE (incremental) | 0.010 |
> | ICAE (one-shot) | 0.004 |
> | **Ours** | **0.116** |
>
> All methods use the same backbone (Llama-2-Chat-7B), so LongMemEval here serves as a **controlled test of C-DIC’s contribution** to long-context QA. Under this setting, C-DIC improves exact-answer accuracy over Full prompting while using substantially fewer input tokens. It clearly emphasizes the effectiveness of our novel C-DIC approach beyond reference-overlap metrics to QA-style correctness. Qualitative examples on LongMemEval are provided in Appendix O (Figure 5 and 6). We also note that our setup does not include LongMemEval-specific enhancements (e.g., explicit temporal signal modeling) and uses a smaller backbone; combining C-DIC with stronger backbones and/or light task adaptation is a natural next step and may further improve QA performance.
>
> In sum, both quantitative and qualitative evaluation results above-mentioned clearly demonstrate the effectiveness of our proposed C-DIC approach on the long-context QA benchmark.
>
> - [1] *LongMemEval: Benchmarking Chat Assistants on Long-Term Interactive Memory (ICLR 2025)*

---

### Official Review · Reviewer_cWNR · 2025-10-30

**Soundness:** 3
**Presentation:** 3
**Contribution:** 3
**Rating:** 6
**Confidence:** 4

**Summary:**

To address the challenges in multi-turn dialogue, such as inefficiencies in processing full dialogue history, static compression/truncation, and poor fidelity, this study introduces the Context-Driven Incremental Compression (C-DIC) framework. This framework conceptualizes dialogue as interleaved contextual threads and maintains a compact memory that stores revisable thread-level compression states. In each dialogue turn, the memory is updated through a lightweight cycle of "retrieval → revise → write-back". For model training, retrieval-aware Truncated Back-propagation Through Time (TBPTT) is employed to reduce computational costs. Validation on the Multi-Session Chat (MSC) dataset and in the zero-shot setting on the REALTALK dataset demonstrates that C-DIC outperforms baseline methods in metrics such as Perplexity (PPL) and BLEU. Additionally, its latency remains stable at 3–3.5 seconds, and it is the only method capable of supporting up to 428 dialogue turns, achieving a balance between efficiency and coherence.

**Strengths:**

1.Balancing Efficiency and Fidelity: The use of threaded memory and incremental compression avoids redundant full-history encoding. The retrieval mechanism focuses on the context relevant to the current dialogue, ensuring stable performance in multi-turn dialogues. While static compression leads to a 1900% increase in Perplexity (PPL), C-DIC achieves a 70% reduction in PPL.

2.Innovative Cross-Turn Mechanism: The core process of "retrieval → revise → write-back" supports dynamic memory updates, addressing the issues of "non-revisable" and "cumulative errors" in static compression, and adapts to the dynamic evolution of dialogues.

3.Efficient Training and Inference: Memory updates during inference are gradient-free, ensuring that latency does not increase with dialogue length. During training, retrieval-aware TBPTT only backpropagates gradients along effective threads, avoiding the computational burden of full-history calculations.

**Weaknesses:**

1.Balancing Efficiency and Fidelity: The use of threaded memory and incremental compression avoids redundant full-history encoding. The retrieval mechanism focuses on the context relevant to the current dialogue, ensuring stable performance in multi-turn dialogues. While static compression leads to a 1900% increase in Perplexity (PPL), C-DIC achieves a 70% reduction in PPL.

2.Innovative Cross-Turn Mechanism: The core process of "retrieval → revise → write-back" supports dynamic memory updates, addressing the issues of "non-revisable" and "cumulative errors" in static compression, and adapts to the dynamic evolution of dialogues.

3.Efficient Training and Inference: Memory updates during inference are gradient-free, ensuring that latency does not increase with dialogue length. During training, retrieval-aware TBPTT only backpropagates gradients along effective threads, avoiding the computational burden of full-history calculations.

Weakness:
1.Methodology: The framework focuses on optimizing the compressor and compression tokens while keeping the response generator fixed. However, the rationale for this design choice is not thoroughly justified. The study relies on initializing the compressor with a pretrained checkpoint from ICAE (Ge et al., 2024) to support this approach, which results in a lack of comprehensive logical grounding.

2.Experiments: The datasets employed are limited to two daily chitchat corpora—Multi-Session Chat (MSC) and REALTALK—without incorporating domain-specific or multilingual data, leaving the model's generalization capabilities untested. Furthermore, the study utilizes a pretrained ICAE checkpoint for initializing the compressor but does not adequately explain how this initialization adapts to multi-turn dialogue scenarios or specify the criteria for parameter tuning, which affects the reproducibility of the results.

3.Conclusion: The paper does not address the inherent limitations of the C-DIC framework, such as its performance boundaries in extremely long dialogues, nor does it discuss potential future applications, resulting in a lack of a forward-looking perspective.

4.Appendix: The hyperparameters, such as the retrieval threshold (τ = 0.8) and decay rate (α = 0.05), are set as fixed values without a strategy for dynamic adjustment. There is no mathematical or experimental justification provided for these settings, and the study does not clarify whether manual optimization of these hyperparameters is necessary, which undermines the scientific rigor of the design.

**Questions:**

see the weakness part

---

> ### Author Response · Authors · 2025-11-21
> **Response to Reviewer cWNR**
>
> Thank you for your detailed review and for highlighting the strengths of C-DIC in terms of efficiency–fidelity trade-offs and the innovative cross turn mechanism. Below we address the raised concerns point by point.
>
> **Q1.1: Methodology: What's the rationale for keeping the response generator frozen?**
>
> **A1.1:** We fix the response generator for two reasons:
>
> - **Isolating the effect of dialogue-level memory.** Our primary goal is to study *incremental compression and thread-aware memory* for multi-turn dialogue, not to re-train or redesign the base LLM. Keeping the generator fixed allows us to attribute performance changes directly to the memory mechanism (threaded memory + incremental compression) rather than to confounding changes in the backbone model.
> - **Practicality and compatibility.** In realistic deployment, the generator is often a large proprietary or heavily tuned model. Our design treats C-DIC as a *drop-in memory layer* that can be attached to such generators without full re-training. This mirrors adapter-style or retrieval-augmented designs that modify external components while reusing a strong frozen LLM.
>
> In the revision, we will clarify that the backbone generator is frozen by design to isolate and highlight the memory contribution.
>
> **Q1.2: Methodology: Initializing the compressor with a pretrained checkpoint from ICAE lacks comprehensive logical grounding.**
>
> **A1.2:** Our goal in this work is to **extend ICAE from one-shot document compression to incremental compression for multi-turn dialogue**, not to re-design the underlying compressor from scratch. Initializing the compressor with the publicly released ICAE checkpoint serves two purposes:
> - **Strong starting point for compression.**
> ICAE has already been trained on large-scale corpora to learn high-quality latent representations for long texts. Using this checkpoint ensures that, before any dialogue-specific training, the model already encodes rich semantic information rather than starting from random weights.
> - **Isolating the effect of incremental, thread-aware memory.** We freeze the response generator and *fine-tune* the ICAE-based compressor. This lets us attribute performance gains to the proposed memory mechanism (threaded, revisable compression and retrieval) rather than to arbitrary differences in base LM or compressor capacity.
>
> Training a brand-new compressor from scratch for each backbone would require massive extra compute and would obscure whether improvements come from the compression architecture or from additional pretraining. We will clarify this in our revision.
>
> To further justify this initialization, we compared (i) starting from the ICAE checkpoint and (ii) a *lightly pretrained* compressor on MSC (using the same ICAE objectives: one-shot continuation and auto-encoding), followed in both cases by our C-DIC training. Evaluated on MSC (5-session setting), we obtain:
>
> | Initialization of Compressor |    PPL    |   BLEU    |    R-L    |    R-1    |    R-2    |
> | :--------------------------: | :-------: | :-------: | :-------: | :-------: | :-------: |
> |       ICAE checkpoint        | **8.427** | **0.030** | **0.160** | **0.206** | **0.040** |
> |   Light pretraining on MSC   |   8.582   |   0.023   |   0.155   |   0.200   |   0.034   |
>
> The ICAE-initialized compressor consistently outperforms the lightly pretrained variant across all metrics, supporting our choice to build on the original ICAE checkpoint as a strong and well-grounded initialization.

---

> > ### Author Response · Authors · 2025-11-21
> > **Response to Reviewer cWNR**
> >
> > **Q2: Experiments**
> >
> > **Q2.1: Generalization capabilities on domain-specific or multilingual dataset**
> >
> > **A2.1:** Our primary focus in this work is evaluating models in *long-context dialogue* settings.
> >
> > Our experiments intentionally focus on long-context daily chit-chat (MSC and REALTALK) because our primary goal is to study incremental compression and dialogue-level memory in naturally long, multi-session conversations with topic drift. MSC contains human–human conversations spanning up to five sessions; we use the official training split with 1,001 episodes (53.3 utterances on average) and evaluate on sessions 2–5, which average 66 utterances per conversation. REALTALK is an even more challenging WhatsApp-style corpus with 10 conversations collected over 21 days, averaging 21.9 sessions and 894.4 utterances per conversation. This setting is precisely where repeated retrieve–compress–write-back operations are critical and where static, one-shot compressors tend to collapse, so it is the most relevant benchmark for our problem formulation.
> >
> > Although both corpora are “daily” dialogues, they already induce a non-trivial shift in style, length, and noisiness: C-DIC is trained only on MSC yet generalizes robustly to REALTALK in a zero-shot setting, despite its much longer and more open-domain nature. This provides concrete evidence that our framework is not tied to a single dataset. In practice, it can be paired with domain- or language-specific backbones and training data without changing the core algorithm. A comprehensive evaluation on multilingual and specialized domains is therefore a natural extension of our work rather than a restriction of the method itself. We will explicitly highlight a systematic multilingual and domain-specific evaluation as important future work in the revised manuscript.
> >
> > **Q2.2: Tuning details of ICAE checkpoint for reproducibility**
> >
> > **A2.2:** In our experiments, we start from the publicly released ICAE checkpoint (using the same architecture and hyperparameters as in Ge et al., 2024) and do **not** modify the compressor architecture. We then fine-tune this checkpoint jointly with C-DIC on the MSC training split using the objectives described in our paper (Equations 7 and 8). The optimization settings for this fine-tuning, including all hyperparameters, are provided in Appendix B. This clarification should make it straightforward for others to reproduce our initialization and tuning procedure.
> >
> > **Q3: Conclusion: Missing limitations and potential future applications of C-DIC**
> >
> > **A3:** We appreciate this comment and agree that the current conclusion does not sufficiently articulate the limitations and forward-looking aspects of C-DIC. In terms of limitations, we currently test C-DIC only on English chit-chat datasets (MSC and REALTALK) with conversations up to roughly 400 turns. We leave systematic evaluation on domain-specific, multilingual, and even longer-term dialogue scenarios as important future work. We will include this point in Conclusion section.
> >
> > **Q4: Appendix: The hyperparameters (the retrieval threshold $\tau$ and decay rate $\alpha$) are set as fixed values without a strategy for dynamic adjustment. Experimental justification provided for these settings is needed. Clarify whether manual optimization of these hyperparameters is necessary**.
> >
> > **A4:** In C-DIC, $\tau$ and $\alpha$ are not directly optimized but interacts with *learned* representations: during training, the encoder and memory updater adapt so that cosine similarities between the query and relevant memory states become compatible with the chosen decision boundary $\tau$ and decay rate $\alpha$.
> >
> > For $\tau$, we swept values from 0.1 to 0.9 on MSC. Empirically, we find that performance is stable for a broad range of values (roughly 0.2–0.8); only very low thresholds (which make almost all states “similar”) or very high thresholds (which make almost no states “similar”) lead to noticeable degradation, because the model either over-updates or under-utilizes memory.
> > Evaluated on MSC (5-session setting), we obtain:
> >
> > | $\tau$ |  PPL   | BLEU  |  R-L  |  R-1  |  R-2  |
> > | :----: | :----: | :---: | :---: | :---: | :---: |
> > |  0.1   | 9.593  | 0.022 | 0.150 | 0.193 | 0.033 |
> > |  0.2   | 8.351  | 0.025 | 0.155 | 0.203 | 0.034 |
> > |  0.3   | 8.334  | 0.027 | 0.162 | 0.212 | 0.041 |
> > |  0.4   | 8.346  | 0.028 | 0.159 | 0.204 | 0.039 |
> > |  0.5   | 8.362  | 0.026 | 0.156 | 0.203 | 0.036 |
> > |  0.6   | 8.325  | 0.028 | 0.157 | 0.208 | 0.037 |
> > |  0.7   | 8.383  | 0.027 | 0.156 | 0.207 | 0.036 |
> > |  0.8   | 8.427  | 0.030 | 0.160 | 0.206 | 0.040 |
> > |  0.9   | 12.202 | 0.022 | 0.139 | 0.190 | 0.027 |
> >
> > This result shows that (i) a coarse grid search over $\tau$ is sufficient, (ii) the method is robust across a wide band of values, and (iii) sophisticated dynamic adjustment is not necessary for good performance in our setting. We will include these tables and discussion in the appendix to make our hyperparameter choices explicit.

---

### Official Review · Reviewer_ebdh · 2025-11-01

**Soundness:** 2
**Presentation:** 2
**Contribution:** 2
**Rating:** 2
**Confidence:** 3

**Summary:**

This paper addresses the efficiency and coherence challenges of multi-turn dialogue by proposing Context-Driven Incremental Compression (C-DIC), which treats conversations as interleaved contextual threads and maintains a compact memory of revisable per-thread compressed states. The key contributions are: (1) demonstrating that static compression methods degrade under multi-turn dynamics with perplexity increasing approximately 1900% from single-turn to multi-turn evaluation, (2) introducing a retrieve→compress→write-back framework where a similarity-based retrieval mechanism fetches relevant thread states, a compressor produces updated states conditioned on retrieved context, and a gradient-free policy either inserts new threads or revises existing ones based on a similarity threshold, and (3) proposing retrieval-aware truncated BPTT that backpropagates gradients only through actually-retrieved memory states. Experiments on MSC and REALTALK datasets show C-DIC achieves perplexity of 8.43 (vs. 27.7 for ICAE one-shot) on MSC and 9.79 (vs. 21.4) on REALTALK, while maintaining approximately 3-3.5 second latency up to 428 turns. The method is evaluated using perplexity, BLEU, and ROUGE metrics, with ablations demonstrating the importance of incremental compression, retrieval-aware TBPTT, and memory-based context threading.

**Strengths:**

### 1. Addresses an important practical problem with clear problem formulation
The paper tackles a genuine challenge in deploying conversational AI systems: managing computational costs and maintaining coherence across long multi-turn interactions. The problem formulation is well-articulated---treating dialogue as interleaved contextual threads requiring revisable memory rather than static compression---and represents a thoughtful reframing of multi-turn challenges. Figure 1 provides suggestive evidence that existing static compression methods may struggle when applied repeatedly across conversation turns, with perplexity increasing sharply after 3-4 consecutive compressions. While this uses the same metrics that limit the paper's evaluation overall, it does motivate investigating alternative approaches designed specifically for conversational dynamics. The paper clearly explains both efficiency concerns (quadratic costs from re-encoding) and coherence concerns (semantic drift over long conversations), establishing why this research direction matters for practical deployment.

### 2. Demonstrates concrete efficiency gains with strong scalability
The efficiency results are well-measured and represent genuine practical contributions. C-DIC maintains approximately constant 3-3.5 second latency regardless of conversation length up to 428 turns, while baseline methods experience out-of-memory errors at 20-40 turns (Figure 3, Table 3). At 30 turns, C-DIC is roughly 2.4× faster than full prompting, with negligible compression overhead. The system is the only method tested that successfully handles 428-turn conversations. The ablation study (Table 2) systematically validates component importance: removing incremental compression causes significant degradation (PPL 9.4→25.5), confirming core design choices matter. Zero-shot transfer to REALTALK (trained on MSC, tested on much longer WhatsApp conversations) suggests the approach may generalize across domains and lengths. While the evaluation has limitations in validating context tracking claims, the efficiency and scalability results are concrete, reproducible, and address real deployment constraints for conversational systems.

**Weaknesses:**

### 1. Evaluation does not validate claimed context tracking capabilities
The paper evaluates using perplexity, BLEU, and ROUGE, which measure how well the model matches human reference responses under teacher-forcing (at each turn, the model generates a response, metrics compare it to the reference, then evaluation continues with the human's actual next turn). These metrics could indirectly validate context tracking if the reference responses require retrieving and using information from distant turns. However, the paper provides no analysis demonstrating that the datasets actually demand this capability.

**Missing dataset characterization**: We don't know (1) what percentage of responses require long-range context (>10 turns back) vs. local context (last 2-3 turns), (2) whether the references could be matched with generic conversational responses that don't require true thread tracking, or (3) the distribution of context dependencies across the 66-utterance (MSC) and 894-utterance (REALTALK) conversations. Without this characterization, strong metric performance could reflect either genuine context tracking or simply learning conversational patterns without the claimed thread awareness.

**Teacher-forcing limitations**: The evaluation never tests whether the model's outputs remain coherent when deployed in closed-loop (where the model must respond to its own previous generations rather than human responses). This masks potential error compounding over long conversations, which is critical for validating the claimed stability across 400+ turns.

**No direct measurement of core claims**: The paper claims "thread awareness" and "contextual fluency" but never directly measures whether the system retrieves correct thread states or uses planted information appropriately. The qualitative example (Figure 6) shows the model making a factual error about which food caused an injury, suggesting the system can fail at context tracking even when generating fluent responses.

The paper should either: (1) provide dataset analysis showing that references require the claimed long-range context tracking (via annotation studies quantifying context dependencies), or (2) add controlled synthetic evaluations where context requirements are explicit (e.g., needle-in-haystack tests where facts planted at turn 5 must be retrieved at turn 50, measuring retrieval precision/recall directly). Also consider LLM-as-judge evaluation and/or human preference studies.

### 2. Technical contribution lacks rigor and key design choices are inadequately explored
Beyond the evaluation issues, the paper's technical contribution is primarily system engineering (combining existing components: ICAE compressor, cosine similarity retrieval, truncated BPTT) without theoretical analysis or comprehensive empirical validation of design choices. Critical hyperparameters lack justification: threshold τ=0.8 shows high sensitivity (Table 4: τ=0.85 increases PPL to 10.1) but no principled selection method is provided; 128 compression tokens are never ablated despite being a capacity bottleneck; the replacement-based write-back policy (Equation 6) completely discards old states without comparison to alternatives like exponential moving average or gating mechanisms. The retrieval-aware TBPTT is presented as a contribution but lacks gradient flow analysis, convergence proofs, or comparison to standard fixed-window TBPTT.

Additionally, the experimental scope is limited: no statistical significance testing across multiple runs, evaluation only on Llama-2-7B without validation on modern models, REALTALK has only 10 conversations, and the related work section inadequately positions the contribution relative to memory-augmented methods (Compressive Transformers, RMT). The catastrophic failure of ICAE (incremental) at PPL 513 is used as motivation but never investigated---is this a fundamental limitation or implementation issue? While these issues are individually addressable, they collectively indicate that the work requires substantial additional development before publication.

**Questions:**

1. Why was there no direct evaluation of context tracking and retrieval accuracy?

2. How were critical hyperparameters selected, and have you compared alternative design choices?

---

> ### Author Response · Authors · 2025-11-21
> **Response to Reviewer ebdh**
>
> Thank you for your valuable feedback and suggestions. We sincerely appreciate your recognition that our method both reframes multi-turn dialogue compression as a revisable, thread-aware memory mechanism and delivers substantial practical benefits in efficiency and scalability. Our detailed responses to your concerns are as follows.
>
> **Q1: Evaluation does not validate claimed context tracking capabilities.**
>
> We thank the reviewer for this detailed analysis of our evaluation setup. Below we respond point-by-point.
>
> **Q1.1: Provide dataset characterization or controlled synthetic evaluations that measure the retrieval performance.**
>
> **A1.1:** We agree that the metrics we report (PPL/BLEU/ROUGE) validate context tracking indirectly. However, it is fundamentally difficult to propose a direct retrieval metric for evaluation. This is because, in C-DIC, contexts are **automatically distributed across memory slots via incremental compression and updates**: a single turn can influence multiple slots, and a given slot can later be reused to update other slots. As a result, there is no simple one-to-one alignment between “turn” and “memory slot”, and it is non-trivial to define a binary notion of a “correctly retrieved slot”.
>
> After thinking deeply to address the reviewer’s concern, we are augmenting our evaluation with a dataset-level analysis that measures how distant the supporting turns are for each response (e.g., the distribution of distances to the earliest turn containing relevant information). This will clarify how often the dataset responses depend on long-range (>10-turn) versus short-range context. We will include these statistics in the revised version. It would be greatly helpful if the reviewer could suggest the metric to test, so that we will turn to add the experiment as soon as possible.
>
> **Q1.2: Teacher-forcing limitations.**
>
> **A1.2:** We agree that closed-loop evaluation is important for assessing long-horizon stability. Following the reviewer’s suggestion, we are working on a closed-loop evaluation where models are conditioned on their generated responses in previous turns.
>
> **Q1.3: No direct measurement of context tracking capabilities and factual error problem**.
>
> **A1.3:** As discussed in A1.1, directly measuring context tracking at the level of individual memory slots is non-trivial in C-DIC, because context is incrementally compressed and spread across slots, and a single turn can influence multiple slots (and vice versa). Instead, we are currently working on a dataset-level analysis that quantifies how often responses depend on long-range context (e.g., by estimating the distance to the earliest supporting turn). We plan to include a summary of this analysis in the revised version.
>
> Regarding the factual error in Figure 6, we appreciate the reviewer highlighting this example.
>
> Interestingly, this particular failure is more likely to be  a *hallucination* of the underlying dialogue model rather than a failure of the memory mechanism: the relevant prior turn is available in the compressed memory, but the generator occasionally produces an incorrect causal attribution. To help better demonstrate our idea, we provide accurate qualitative examples in Appendix O (Figure 5 and 6).
>
> **Q1.4: Consider LLM-as-judge evaluation and/or human preference studies.**
>
> **A1.4:** We appreciate the suggestion and agree that both LLM-as-judge and human preference studies are valuable for assessing dialogue quality. However, in our setting they are non-trivial to apply reliably.
>
> For LLM-as-judge evaluation, our preliminary experiments indicate that even strong LLMs struggle to robustly assess conversations spanning hundreds of turns. Human evaluation faces a similar challenge at this scale: asking annotators to read and reliably reason over entire ~400-turn conversations is extremely demanding and costly, and simple truncation undermines the long-context focus of our work. Designing a high-quality, hierarchical human evaluation protocol for such long dialogues is therefore beyond the scope of this submission. That said, we agree that richer human and LLM-based evaluations are an important next step. We will mention this explicitly as a limitation and as a key direction for future work.

---

> > ### Author Response · Authors · 2025-12-03
> > **A1.2 Closed-Loop Evaluation**
> >
> > **Q1.2: Teacher-forcing limitations — does the model remain stable in closed-loop over long dialogues?**
> >
> > We thank the reviewer for raising a deployment-related concern: **errors can compound** when a model must condition on its *own* previous generations, potentially harming coherence over long horizons. To make the discussion precise, we first clarify the two evaluation settings:
> >
> > - **Teacher forcing:**  at turn t, the model generates the assistant response conditioned on the **ground-truth** dialogue history (turns $1 \ldots t−1$). This is **standard** for reference-based metrics (PPL/BLEU/ROUGE) because it evaluates responses under a **fixed** context.
> > - **Closed-loop:** at turn t, the model conditions on the same user query for all turns, but the assistant-side history is replaced by the model’s **own generated** responses from earlier turns.
> >
> > We also note that *closed-loop evaluation on offline dialogue corpora is not a perfect proxy for real deployment*: because future user turns are taken from the dataset, they may become partially misaligned with the earlier model generations. Thus, this is **error-amplifying** setting rather than a fully realistic interaction. Nonetheless, this provides a useful **stress test** for long-horizon stability. To directly address the reviewer’s concern, we add two closed-loop evaluations:
> >
> > ### **(1) REALTALK (per-session): closed-loop comparison across methods**
> >
> > We evaluate all methods (baselines + ours) under closed-loop in the REALTALK *per-session* setting. This setting keeps most baselines runnable and enables a meaningful comparison, since several compression methods exceed GPU memory under the full long-context (all-sessions) configuration. ICAE(append) remains OOM even per-session, which we report explicitly. Importantly, per-session closed-loop still captures **within-session error accumulation**.
> >
> > | Models | PPL ↓ | BLEU ↑ | R-L ↑ | R-1 ↑ | R-2 ↑ |
> > | --- | --- | --- | --- | --- | --- |
> > | Full prompting | 29.666 | 0.020 | 0.106 | 0.150 | 0.017 |
> > | Truncation | 28.174 | 0.021 | 0.109 | 0.156 | 0.019 |
> > | Summarization | 27.977 | 0.022 | 0.110 | 0.162 | 0.021 |
> > | In-Session RAG | 26.789 | 0.020 | 0.103 | 0.149 | 0.015 |
> > | AutoCompressor | 13.111 | 0.020 | 0.111 | 0.144 | 0.021 |
> > | ICAE (incremental) | 124.024 | 0.020 | 0.068 | 0.088 | 0.012 |
> > | ICAE (one-shot) | 17.364 | 0.024 | 0.109 | 0.152 | 0.023 |
> > | ICAE (append) | out-of-memory | - | - | - | - |
> > | Ours | **9.754** | **0.034** | **0.133** | **0.173** | **0.031** |
> >
> > Under this closed-loop evaluation, our method remains the strongest overall among runnable baselines, achieving **the best performance** across all reported metrics. Scores are slightly lower than the teacher-forcing results in Table 1, which is expected under closed-loop evaluation due to compounding generation errors and user–assistant misalignment. Despite this **error-amplifying** setup, our method exhibits only a small gap between teacher forcing and closed-loop evaluation, **suggesting that C-DIC is stable under self-conditioned generation.**
> >
> > ### **(2) REALTALK (all-sessions): closed-loop vs. teacher-forcing for our method**
> >
> > To probe long-horizon behavior directly, we additionally evaluate our method on REALTALK in the *all-sessions* setting (full multi-session history; 400+ turns) under both teacher forcing and closed-loop:
> >
> > | Model | PPL ↓ | BLEU ↑ | R-L ↑ | R-1 ↑ | R-2 ↑ |
> > | --- | --- | --- | --- | --- | --- |
> > | Ours (teacher forcing) | **9.556** | **0.036** | **0.140** | **0.184** | **0.035** |
> > | Ours (closed-loop) | 9.576 | **0.036** | 0.137 | 0.182 | 0.032 |
> >
> > Performance remains strong in the *all-sessions* setting, indicating that our compressed memory states remain useful across **hundreds of turns**. Moreover, the closed-loop scores are very close to teacher forcing, indicating **limited error accumulation** even over long histories.
> >
> > These closed-loop experiments show that our improvements are not an artifact of teacher forcing and provide direct evidence that C-DIC remains stable when conditioning on its own generations over long conversations, addressing the reviewer’s concern.

---

> ### Author Response · Authors · 2025-11-21
> **Response to Reviewer ebdh**
>
> **Q2: Technical contribution lacks rigor and key design choices are inadequately explored.**
>
> We appreciate the reviewer’s detailed comments on the technical aspects of our design. We would claim that our novelty lies in (i) our design that maintains *revisable contextual threads* over multi-turn dialogue, (ii) the incremental compression and write-back policy that enables long-horizon operation, and (iii) the retrieval-aware TBPTT scheme tailored to dialogue-level memory rather than contiguous token sequences. For the specific design choices, we respond to each concern below, using the MSC session-5 split (the longest session) for all evaluations.
>
> **Q2.2: Sensitivity and lack of justification for the similarity threshold $\tau$.**
>
> **A2.2:** We agree that the detailed description of the similarity threshold  $\tau$ is required. In C-DIC,  $\tau$ is not directly optimized but interacts with *learned* representations: during training, the encoder and memory updater adapt such that cosine similarities between the query and relevant memory states evolve in a way that is compatible with a fixed decision boundary $\tau$.
>
> Empirically, we find that performance is stable for a broad range of values (roughly 0.2–0.8); only very low thresholds (which make almost all states “similar”) or very high thresholds (which make almost no states “similar”) lead to noticeable degradation, because the model either over-updates or under-utilizes memory. The results are as follows:
>
> | $\tau$ |  PPL   | BLEU  |  R-L  |  R-1  |  R-2  |
> | :----: | :----: | :---: | :---: | :---: | :---: |
> |  0.1   | 9.593  | 0.022 | 0.150 | 0.193 | 0.033 |
> |  0.2   | 8.351  | 0.025 | 0.155 | 0.203 | 0.034 |
> |  0.3   | 8.334  | 0.027 | 0.162 | 0.212 | 0.041 |
> |  0.4   | 8.346  | 0.028 | 0.159 | 0.204 | 0.039 |
> |  0.5   | 8.362  | 0.026 | 0.156 | 0.203 | 0.036 |
> |  0.6   | 8.325  | 0.028 | 0.157 | 0.208 | 0.037 |
> |  0.7   | 8.383  | 0.027 | 0.156 | 0.207 | 0.036 |
> |  0.8   | 8.427  | 0.030 | 0.160 | 0.206 | 0.040 |
> |  0.9   | 12.202 | 0.022 | 0.139 | 0.190 | 0.027 |
>
> In the revision, we will report this extended sensitivity analysis where we sweep $\tau$ over a wide range, and show that within the recommended band the impact on performance is modest.
>
> **Q2.3: Missing ablation on compression capacity (128 tokens).**
>
> **A2.3:** Our goal is to study incremental compression for multi-turn dialogue by extending ICAE, rather than to re-optimize the underlying compressor for static documents. In the main paper we therefore used the publicly released ICAE checkpoint, which is trained on large corpora and only provided for a single compression size (128 tokens).
>
> We agree, however, that the number of compression tokens can matter in the multi-turn setting.
> To evaluate this, we trained variants of the compressor with 64, 128, and 256 tokens on MSC using the same ICAE objectives (one-shot continuation and auto-encoding), and then fine-tuned the dialogue model with our method. The results on MSC are:
>
> | Compression token length |  PPL  | BLEU  |  R-L  |  R-1  |  R-2  |
> | :----------------------: | :---: | :---: | :---: | :---: | :---: |
> |            64            | 8.604 | 0.023 | 0.157 | 0.201 | 0.036 |
> |           128            | 8.582 | 0.023 | 0.155 | 0.200 | 0.034 |
> |           256            | 8.646 | 0.022 | 0.160 | 0.205 | 0.037 |
>
> These results indicate that performance is relatively stable across 64–256 tokens, with differences in PPL and generation metrics being modest and without a clear monotonic trend. This suggests that C-DIC is not overly sensitive to the exact compression capacity within this range.
>
>
> **Q2.4: Replacement-based write-back policy vs Exponential Moving Average (EMA) or gating mechanism.**
>
> **A2.4:** We implemented two alternatives to the simple replacement-based write-back: (i) EMA updates with decay factors  $\beta \in \{0.3, 0.5, 0.7\}$, using $\beta \cdot \text{memory}\_\text{old} + (1-\beta) \cdot \text{memory}\_\text{new}$, and (ii) a 2-layer gating network that learns to interpolate between the old and new memory states. As shown in the table below, EMA brings at best marginal gains only on the R-1 metric and often yields noticeably worse performance on other metrics, while the gated variant provides only small improvements on some ROUGE scores at the cost of additional complexity.
>
> Given this trade-off, we adopt the replacement policy as the simpler and more robust choice.
>
> | Memory Update |    PPL    |   BLEU    |    R-L    |    R-1    |    R-2    |
> | ------------- | :-------: | :-------: | :-------: | :-------: | :-------: |
> | Replacement   | **8.427** | **0.030** |   0.160   |   0.206   | **0.040** |
> | EMA (0.3)     |   8.442   |   0.027   |   0.157   |   0.208   |   0.035   |
> | EMA (0.5)     |   8.836   |   0.027   |   0.155   |   0.203   |   0.036   |
> | EMA (0.7)     |   9.929   |   0.021   |   0.145   |   0.186   |   0.030   |
> | Gate          |   8.503   |   0.026   | **0.162** | **0.209** |   0.037   |

---

> ### Author Response · Authors · 2025-11-21
> **esponse to Reviewer ebdh**
>
> **Q2.5: Retrieval-aware TBPTT lacks gradient flow analysis, convergence proofs, or comparison to standard fixed-window TBPTT.**
>
> **A2.5:** We would like to clarify that the standard fixed-window TBPTT broken the gradient flow through selected memory states.
>
> Specifically, using the notation in Section 3.3, we minimize the per-turn NLL as described in Equation 7:
> $$
> \begin{equation}
>     L = \frac{1}{T}\sum_{t=1}^T \ell_t,
>     \qquad
>     \ell_t = - \log P_\phi(r_t \mid q_t, R_t),
> \end{equation}
> $$
> where $R_t \subset M_{<t}$ is the retrieved set and each slot $Z_s \in M_{<t}$ is a compressed memory state.
>
> Under full BPTT, the gradient with respect to a memory slot $Z_s$ aggregates contributions from all future turns where that slot is actually consulted (i.e., $Z_s \in R_t$):
> $$
> \begin{equation}
>     \frac{\partial L}{\partial Z_s}
>     =
>     \sum_{t=1}^T \mathbf{1}[Z_s \in R_t]\,
>     \frac{\partial \ell_t}{\partial Z_s}.
> \end{equation}
> $$
> However, implementing full BPTT requires keeping the computation graph for all $T$ turns in memory, so the activation cost grows linearly with dialogue length. For long conversations this becomes prohibitive in practice and quickly leads to out-of-memory (OOM) errors.
>
> In standard fixed-window TBPTT with horizon $K$, at each turn $t$ all slots older than $K$ steps are detached from the computation graph before retrieval. Concretely, retrieval reads
>
> $$
> \begin{equation}
>     \tilde Z_s =
>     \begin{cases}
>         \operatorname{stopgrad}(Z_s), & s \le t-K, \\
>         Z_s, & s > t-K,
>     \end{cases}
>     \qquad
>     R_t \subset \{\tilde Z_s\}_{s < t}.
> \end{equation}
> $$
>
> By the chain rule, for any $s \le t-K$,
> $$
> \begin{equation}
>     \frac{\partial \ell_t}{\partial Z_s}
>     =
>     \frac{\partial \ell_t}{\partial \tilde Z_s}\,
>     \frac{\partial \tilde Z_s}{\partial Z_s}
>     =
>     \frac{\partial \ell_t}{\partial \tilde Z_s}\cdot 0
>     = 0,
> \end{equation}
> $$
>
> even if $Z_s \in R_t$ (i.e., the slot is selected and used at turn $t$). The truncated gradient therefore becomes
>
> $$
> \begin{equation}
>     \frac{\partial L}{\partial Z_s}
>     =
>     \sum_{t=1}^T \mathbf{1}[Z_s \in R_t]\,
>     \mathbf{1}[t - s < K]\,
>     \frac{\partial \ell_t}{\partial Z_s},
> \end{equation}
> $$
>
> so any selected memory state $Z_s$ that is retrieved only after it falls outside the $K$-step window receives no gradient signal from those distant uses.
>
> **Q2.6**: **Limited experimental scope**
>
> We appreciate the reviewer’s concerns about the breadth and robustness of our empirical evaluation. Below we address each point.
>
> **Q2.6.1: No significance testing across multiple runs**.
>
> **A2.6.1:** We agree that reporting variance across runs would strengthen the results. We are currently running additional seeds on MSC (for the main baselines and C-DIC) and will report mean and standard deviation of PPL / BLEU / ROUGE in the revised version.
>
> **Q2.6.2:Evaluation only on Llama-2-7B.**
>
> **A2.6.2:** Our goal in this work is to adapt the state-of-the-art one-shot compression model ICAE to the multi-turn dialogue setting via our novel C-DIC incremental compression framework. Accordingly, we build on the publicly available ICAE checkpoint, which is trained with a Llama-2-7B backbone. Training ICAE-style compressors for additional pedestal models is non-trivial and computationally expensive and is orthogonal to our main contribution: designing and evaluating a general incremental compression mechanism given a pre-trained one-shot compressor. For this reason, we concentrate our experiments on rigorously demonstrating the effectiveness and reliability of C-DIC under multi-turn dialogue setting. While we are not able to pretrain other ICAE variants during the rebuttal period due to this substantial overhead, the proposed C-DIC framework itself is model-agnostic: it only requires access to a compatible one-shot compressor and can therefore be combined with other backbone LMs as such compressors become available. We will make this model-agnostic design choice explicit in the revised manuscript.
>
> **Q2.6.3:REALTALK has only 10 conversations**.
>
> **A2.6.3:** While REALTALK contains only 10 dialogues, each conversation is extremely long (on average 447.1 turns). We evaluate the model turn-by-turn, resulting in 1898 evaluation instances in total. Thus, REALTALK serves as a challenging stress test for very long-horizon dialogue rather than a small-scale dataset.

---

> ### Author Response · Authors · 2025-11-21
> **Response to Reviewer ebdh**
>
> **Q2.6.4: Related work section inadequately positions the contribution relative to memory-augmented methods (Compressive Transformers, RMT)**
>
> **A2.6.4:** We appreciate this comment and agree that the current Related Work section should discuss memory-augmented Transformer architectures such as Compressive Transformers and RMT.
>
> Compressive Transformers and RMT modify the **internal** attention/memory mechanism of the Transformer, introducing compressed token/segment memories inside the model layers to extend effective context length for generic long-sequence modeling. By contrast, C-DIC is an **external dialogue-level memory system**: it organizes multi-turn interactions into revisable contextual threads and feeds a compact state back into an otherwise standard generator. Our contribution lies in this dialogue-level memory abstraction and the retrieval-aware training scheme, rather than in proposing a new Transformer memory cell, and is therefore complementary to memory-augmented architectures. We have updated Related work section on this point.
>
> **Q2.6.5: The catastrophic failure of ICAE (incremental) at PPL 513 is used as motivation but never investigated. Is this a fundamental limitation or implementation issue?**
>
> **A2.6.5:** The ICAE incremental variant fails for structural reasons (latent drift from applying a one-shot objective incrementally), while the ICAE-append (accumulating compressed context) and ICAE-one-shot (re-encode the full available context each turn) variants help at short lengths but OOMs in long contexts.
>
> We would like to emphasize that the failure is structural rather than an implementation issue. ICAE is trained with a **one-shot compression objective**: it learns to encode a contiguous context span into a latent in a single step. In the incremental variant, however, we repeatedly apply this compressor to its *own compressed outputs* as the dialogue progresses. This leads to **latent drift and error compounding**, because the model is never trained to use or update “already-compressed” contexts. Empirically, the response quality degrades rapidly across turns, which manifests as the very high perplexity (PPL ≈ 513) we report.
>
> By contrast, the ICAE-append and ICAE-one-shot baselines perform reasonably well at short and medium lengths, but eventually run out of memory on very long conversations, as shown in Figure 3 and Table 11. In other words, ICAE-append and ICAE-one-shot are effective but not scalable, whereas the naive incremental application is scalable but unstable. This mismatch is exactly what motivates our design: C-DIC modifies the architecture and training scheme to support **incremental** use of compression while mitigating the catastrophic degradation observed in incremental ICAE.
>
> We will mention this discussion in the revised version and explicitly frame the ICAE (incremental) result as evidence that naive reuse of one-shot compressors in an incremental fashion is problematic.

---

> ### Author Response · Authors · 2025-11-27
> **A1.1: Dataset Characterization of Long-Range Dependencies**
>
> **Q1.1: Evaluation does not validate claimed context tracking capabilities. Provide dataset characterization or controlled synthetic evaluations that measure the retrieval performance.**
>
> We thank the reviewer for highlighting that our main automatic metrics (PPL/BLEU/ROUGE) are reference-based and therefore do not directly reflect long-range context tracking by themselves. We address this concern with an analysis of dataset characteristics by quantifying long-range dependencies and response genericity.
>
> **A1.1:**
>
> ### (1) Dataset characterization: do MSC / REALTALK require distant context?
>
> **LLM-based annotation.** We leverage GPT-4o to label whether a *candidate past utterance* contains **necessary or materially helpful** information for producing the **reference assistant response** to a target query. In particular, we feed **pair instances** consisting of: (i) single historical utterance and (ii) the final-turn context (latest user query + reference response), to GPT-4o with an instruction$^1$. We adopt this approach based on our empirical observation that excessively long prompts substantially degrade LLM performance on relevance judgment. Upon producing per-utterance relevance labels, we report **conversation-level** statistics (e.g., whether any supporting utterances (*context*) occur ≥10 turns back).
>
> In addition, we label each target reference response as generic vs. not-generic using a separate instruction.$^2$. A response is generic if it would remain appropriate for a wide range of plausible user queries without relying on specific dialogue content (e.g., greeting, generic acknowledgments or vague continuations), and not-generic otherwise.
>
> **Sampling.** We sample 500 and 320 conversations from MSC and REALTALK, respectively, while restricting to dialogues with ≥11 turns so that “≥10 turns back” is well-defined. These sample sizes provide stable estimation of conversation-level rate at reasonable cost (worst-case 95% margin ≈ ±4–6pp), consistent with prior LLM annotation/judge practice that uses hundreds of samples as a cost-aware yet reliable evaluation scheme [1].
>
> **Results.**
>
> | Dataset | Sampled dialogues | Evidence from ≥10 turns (%) | Dialogues w/ farthest ≥10 (%) | Generic resp. (%) |
> | :--- | :---: | :---: | :---: | :---: |
> | REALTALK | 320 | 66.94 | 40.31 | 6.25 |
> | MSC | 500 | 44.92 | 70.80 | 2.00 |
>
> “Evidence from ≥10 turns (%)” measures the fraction of *supporting* utterances that occur ≥10 turns before the final response, while “Dialogues w/ farthest support ≥10 turns (%)” measures how often the *most distant* supporting utterance occurs ≥10 turns back.
>
> ### (2) Human verification of judge reliability
>
> To validate the above relevance labeling, we run a human verification study with **three annotators** on **50 randomly sampled items per task and dataset**, following the recommended LLM-as-a-judge verification setting in [2]. Annotators independently label (i) helpful-turn vs. not-helpful and (ii) generic vs. not-generic, and we report LLM accuracy against majority vote plus inter-annotator agreement.
>
> | Task |  | **MSC** |  |  | **REALTALK** |  |
> | --- | :---: | :---: | :---: | :---: | :---: | :---: |
> |  | ACC (%) | Agree | Fleiss’ κ | ACC (%) | Agree | Fleiss’ κ |
> | Helpful-turn label | 92.000 | 0.920 | 0.527 | 90.000 | 0.987 | 0.921 |
> | Generic-response label | 95.918 | 0.973 | 0.652 | 96.000 | 0.973 | 0.653 |
>
> The results show high agreement with human judgments (≈90–96% accuracy) and strong inter-annotator consistency, demonstrating the reliability of GPT-4o labels for dataset-level characterization.
>
> Our analysis of dataset characteristics clearly and reliably demonstrates that long-range dependencies are common and that truly generic final responses are rare. In sum, these findings indicate that our benchmarks meaningfully evaluate long-context tracking, rather than being solvable via short-range cues or templated replies.
>
> ###### $^1$ “You are given the latest user query and the assistant’s response of a conversation, along with an utterance from the past conversation. Determine whether the utterance is helpful for generating the assistant’s response to the latest user query. Output ‘helpful’ or ‘not helpful’.”
>
> ###### $^2$ "You are given the latest user query and the assistant's final response of a conversation. Decide whether the assistant’s final response is generic. A response is “generic” if it could be pasted into many different conversations/questions with minimal editing (e.g., greetings/farewells/small talk). Otherwise “not generic”. Output only generic or not generic.”
>
> - [1] Pangakis, Wolken, Fasching. *Automated Annotation with Generative AI Requires Validation.* arXiv:2306.00176 (2023). [arXiv](https://arxiv.org/abs/2306.00176)
> - [2] Calderon, Reichart, Dror. *The Alternative Annotator Test for LLM-as-a-Judge: How to Statistically Justify Replacing Human Annotators with LLMs.* ACL 2025 (arXiv:2501.10970).

---

> ### Author Response · Authors · 2025-12-02
> **A2.6.1 Significance testing across multiple runs**
>
> **Q2.6.1: Limited experimental scope - no significance testing across multiple runs**.
>
> To address robustness across runs, we repeat experiments with **three random seeds** (42/43/44) and report **mean±std**.
>
> ### MSC (mean±std over 3 seeds)
>
> | Model | PPL ↓ | BLEU ↑ | ROUGE-L ↑ | ROUGE-1 ↑ | ROUGE-2 ↑ |
> | --- | --- | --- | --- | --- | --- |
> | AutoCompressor | 9.109 ± 0.273 | 0.014 ± 0.002 | 0.121 ± 0.002 | 0.145 ± 0.003 | 0.021 ± 0.001 |
> | ICAE (incremental) | 561.702 ± 347.397 | 0.007 ± 0.001 | 0.063 ± 0.006 | 0.075 ± 0.008 | 0.005 ± 0.001 |
> | ICAE (one-shot) | 29.188 ± 1.371 | 0.017 ± 0.000 | 0.132 ± 0.001 | 0.188 ± 0.002 | 0.027 ± 0.001 |
> | **Ours** | **8.385 ± 0.042** | **0.025 ± 0.002** | **0.159 ± 0.001** | **0.202 ± 0.003** | **0.037 ± 0.000** |
>
> ### REALTALK (mean±std over 3 seeds; per-session)
>
> | Model | PPL ↓ | BLEU ↑ | ROUGE-L ↑ | ROUGE-1 ↑ | ROUGE-2 ↑ |
> | --- | --- | --- | --- | --- | --- |
> | AutoCompressor | 12.283 ± 0.352 | 0.020 ± 0.001 | 0.090 ± 0.030 | 0.138 ± 0.003 | 0.020 ± 0.001 |
> | ICAE (incremental) | 135.827 ± 39.254 | 0.019 ± 0.001 | 0.071 ± 0.003 | 0.089 ± 0.003 | 0.013 ± 0.001 |
> | ICAE (one-shot) | 25.115 ± 3.388 | 0.025 ± 0.001 | 0.113 ± 0.005 | 0.159 ± 0.006 | 0.024 ± 0.002 |
> | **Ours** | **9.764 ± 0.043** | **0.034 ± 0.001** | **0.136 ± 0.002** | **0.177 ± 0.002** | **0.032 ± 0.001** |
>
> Across three independent runs, our method exhibits **very low variance** (PPL std ≈ 0.04 on both MSC and REALTALK) while maintaining the **same ranking across seeds** and delivering **consistent gains** over the strongest baseline ICAE(one-shot) on every metric reported.
>
> Importantly, our contribution is not limited to metric improvements. On REALTALK, ICAE(append) and ICAE(one-shot) run into **OOM** in the full long-context configuration (Figure 3 and Table 11), whereas our method is specifically designed to make the compression **operate stably over hundreds of turns** (Figure 4). For the main table we therefore report REALTALK in the \emph{per-session} setting to ensure a fair, runnable comparison across methods.
>
> As additional evidence, we follow established guidance for significance testing in NLP [3] and compute p-values from a paired t-test on the seed-wise differences. Against ICAE(one-shot), improvements are statistically significant on **both datasets**(**all p < 0.05)**:
>
> | Dataset | log(PPL) | BLEU | R-L | R-1 | R-2 |
> | --- | --- | --- | --- | --- | --- |
> | MSC | 2.8×10⁻⁴ | 0.010 | 9.6×10⁻⁵ | 0.003 | 5.6×10⁻⁴ |
> | REALTALK | 0.004 | 0.002 | 0.013 | 0.027 | 0.024 |
>
> These results indicate the improvements are not explained by seed variability, directly addressing the reviewer’s request for multi-run robustness and significance testing.
>
> - [3] Dror, Baumer, Shlomov, Reichart. *The Hitchhiker’s Guide to Testing Statistical Significance in Natural Language Processing.* ACL 2018 *(*https://aclanthology.org/P18-1128/*)*

---

### Official Review · Reviewer_DQty · 2025-11-20

**Soundness:** 2
**Presentation:** 3
**Contribution:** 2
**Rating:** 4
**Confidence:** 3

**Summary:**

This paper addresses the challenges of efficiency and fidelity in model context management during multi-turn conversations by proposing the C-DIC framework, which treats dialogues as interleaved context threads. It employs a "retrieval-revision-rewrite" cycle to manage modifiable compressed memory and incorporates retrieval-aware TBPTT for optimized training.

**Strengths:**

1. The results showed that C-DIC outperformed all metrics comprehensively, including PPL, BLEU, and ROUGE, with only 3.36 seconds required to process 428 rounds of dialogue. It also demonstrated strong zero-shot transferability. Ablation studies confirmed the necessity of each component, providing an efficient and scalable solution for long-range dialogue modeling, with notable innovation and practicality.
2. The study validates the effectiveness of the gradient-free memory management strategy in dynamic conversations, achieving a balance between reasoning efficiency and contextual fidelity.

**Weaknesses:**

1. Lack of experiments with multiple pedestal models: The main experiment was only conducted on Llama2-7B, lacking more extensive pedestal models, including experimental results from more advanced models.
2. These experiments were mainly conducted on two daily corpus databases. Can they generalize to a wider range of dialogue data, such as multilingual data, or specific domain dialogues, such as dialogues in the medical field and debates.

**Questions:**

Please refer to the weakness part.

---

> ### Author Response · Authors · 2025-11-21
> **Response to Reviewer DQty**
>
> Thank you for your valuable feedback and suggestions. We sincerely appreciate your recognition of the notable innovation and practicality of our method.
>
> **Q1: The main experiment was only conducted on Llama2-7B, lacking more extensive pedestal models.**
>
> **A1:** Our goal in this work is to adapt the state-of-the-art one-shot compression model ICAE to the multi-turn dialogue setting via our novel C-DIC incremental compression framework. Accordingly, we build on the publicly available ICAE checkpoint, which is trained with a Llama-2-7B backbone. Training ICAE-style compressors for additional pedestal models is non-trivial and computationally expensive and is orthogonal to our main contribution: designing and evaluating a general incremental compression mechanism given a pre-trained one-shot compressor. For this reason, we concentrate our experiments on rigorously demonstrating the effectiveness and reliability of C-DIC under multi-turn dialogue setting. While we are not able to pretrain other ICAE variants during the rebuttal period due to this substantial overhead, the proposed C-DIC framework itself is model-agnostic: it only requires access to a compatible one-shot compressor and can therefore be combined with other backbone LMs as such compressors become available. We will make this model-agnostic design choice explicit in the revised manuscript.
>
>
> **Q2: These experiments were mainly conducted on two daily corpus databases.**
> **Generalize to a wider range of dialogue data, such as multilingual data, or specific domain dialogues, such as dialogues in the medical field and debates.**
>
> **A2:** Our experiments intentionally focus on long-context daily chit-chat (MSC and REALTALK) because our primary goal is to study incremental compression and dialogue-level memory in naturally long, multi-session conversations with re-engagement and topic drift. MSC contains human–human conversations spanning up to five sessions; we use the official training split with 1,001 episodes (53.3 utterances on average) and evaluate on sessions 2–5, which average 66 utterances per conversation. REALTALK is an even more challenging WhatsApp-style corpus with 10 conversations collected over 21 days, averaging 21.9 sessions and 894.4 utterances per conversation. This setting is precisely where repeated retrieve–compress–write-back operations are critical and where static, one-shot compressors tend to collapse, so it is the most relevant benchmark for our problem formulation.
>
> Although both corpora are “daily” dialogues, they already induce a non-trivial shift in style, length, and noisiness: C-DIC is trained only on MSC yet generalizes robustly to REALTALK in a zero-shot setting, despite its much longer and more open-domain nature. This provides concrete evidence that our framework is not tied to a single dataset. Methodologically, C-DIC is model- and domain-agnostic: it operates on dialogue turns and latent memory states via the same retrieve → revise → write-back loop, without assuming any specific domain (chit-chat vs. medical vs. debate) or language. In practice, it can be paired with domain- or language-specific backbones and training data without changing the core algorithm. A comprehensive evaluation on multilingual and specialized domains such as medical consultation or debate is therefore a natural extension of our work rather than a restriction of the method itself. Given that this review was released only one day before the rebuttal deadline, it is unfortunately not realistic to design, run, and carefully validate such additional experiments in this short window, but we will explicitly highlight a systematic multilingual and domain-specific evaluation as important future work in the revised manuscript.

---

### Meta-Review · Area_Chair_oQvt · 2026-01-06

**Summary:**

This submission studies long-horizon multi-turn dialogue context compression and proposes Context-Driven Incremental Compression (C-DIC), which frames a conversation as interleaved contextual threads and maintains a compact, revisable dialogue memory updated via a lightweight retrieve → revise → write-back loop. The paper further introduces retrieval-aware TBPTT to train the memory module efficiently without full-history backpropagation. Overall, the work targets an important practical problem for conversational systems: scaling to long dialogue histories while preserving coherence and controlling computation.
Across reviews, there is broad agreement that the paper’s core idea—treating dialogue memory as revisable thread states and explicitly supporting cross-turn revision—addresses a real failure mode of one-shot/static compressors under repeated use. Empirically, the method demonstrates strong scalability: stable inference latency over very long dialogues (hundreds of turns) where several baselines run into OOM or degrade substantially. The ablation results support that incremental compression, retrieval, and the training scheme each contribute materially. The MSC→REALTALK transfer is also a meaningful stress test for long-horizon conversational dynamics.

**Reviewer Concerns:**

1. The experimental domains, backbones, and language types are relatively limited, raising concerns about the generalizability of the method (DQty, cWNR, QNTu).
2. Key design choices lack ablation experiments (ebdh, cWNR).
3. The metrics used in the experiments do not fully reflect real-world outcomes; direct retrieval correctness metrics (precision/recall) should be added (ebdh).

**Reviewer Scores:**

Reviewer DQty is unlikely to change the score. The main concerns focus on the limited experimental backbones and datasets, and the authors did not effectively add targeted experiments in the rebuttal.
Reviewer ebdh may increase the score to 4 or higher. During the rebuttal, the authors added utterance-level helpfulness and generic-response annotations using GPT-4o, multi-seed experiments, closed-loop evaluation, and ablation experiments, which alleviated some concerns.
Reviewer QNTu and cWNR may maintain their scores or slightly increase them. During the rebuttal, the authors explained issues such as the rationale for freezing the generator, grounding using only ICAE initialization requiring additional fine-tuning, and the lack of coverage of new long-context benchmarks.

---

### Decision · Program_Chairs · 2026-01-26

Reject